# Endocardial HDAC3 is required for myocardial trabeculation

Jihyun Jang [1,2], Mette Bentsen [3], Ye Jun Kim [4], Erick Kim [4], Vidu Garg [1,2], Chen-Leng Cai[5], Mario Looso [3] & Deqiang Li [1,2] ✉

Failure of proper ventricular trabeculation is often associated with congenital heart disease. Support from endocardial cells, including the secretion of extracellular matrix and growth factors is critical for trabeculation. However, it is poorly understood how the secretion of extracellular matrix and growth factors is initiated and regulated by endocardial cells. We find that genetic knockout of histone deacetylase 3 in the endocardium in mice results in early embryo lethality and ventricular hypotrabeculation. Single cell RNA sequencing identifies significant downregulation of extracellular matrix components in histone deacetylase 3 knockout endocardial cells. Secretome from cultured histone deacetylase 3 knockout mouse cardiac endothelial cells lacks transforming growth factor ß3 and shows significantly reduced capacity in stimulating cultured cardiomyocyte proliferation, which is remarkably rescued by transforming growth factor ß3 supplementation. Mechanistically, we identify that histone deacetylase 3 knockout induces transforming growth factor ß3 expression through repressing microRNA-129-5p. Our findings provide insights into the pathogenesis of congenital heart disease and conceptual strategies to promote myocardial regeneration.

Congenital heart disease (CHD) is the most common congenital disorder, and the leading cause of mortality from birth defects[1]. Defective myocardial growth is either the primary manifestation or associated with many forms of CHD[2,3]. In the early primitive heart (mouse embryonic day 8-10), inner cardiac endothelial cells (CECs), also called endocardial cells, and outer cardiomyocytes (CMs), which are separated by extracellular matrix (ECM), co-contribute to early myocardial trabeculation (i.e., formation of myocardial meshwork extending to cardiac chamber)[4–6]. CECs juxtapose myocardial trabeculae and lead myocardial trabeculation by secretion of growth factors such as Neuregulin 1 (NRG1) and transforming growth factor ß (TGFß)[2,3,5]. Meanwhile, CECs support trabeculation by secreting ECM (e.g., various collagens, versican) that facilitate the traveling of CEC secreted growth factors to trabecular myocardium and support trabecular CM

proliferation[4,7]. Surprisingly, disruption of ECM components alone (e.g., hyaluronan synthase-2, versican) prevents the proper formation of trabeculae in mice[8,9]. Although CECs play such important roles during ventricular trabeculation, it remains poorly understood how these ECM proteins and growth factors are regulated in the developing CECs.

TGFß signaling pathway is involved in many fundamental cellular processes in both adults and developing embryos including cell proliferation, differentiation, migration, and apoptosis[10,11]. The TGFß superfamily includes many different ligands (e.g., Bone morphogenetic proteins, Growth and differentiation factors, Activin, Nodal, TGFß). These ligands bind to their specific type II receptors, which triggers phosphorylation of type I receptor, eventually leads to phosphor-Smad2/3 localized to the nucleus and drives a variety of

[1]Center for Cardiovascular Research, Abigail Wexner Research Institute, Nationwide Children's Hospital, Columbus, OH 43215, USA. [2]Department of Pediatrics, The Ohio State University College of Medicine, Columbus, OH 43215, USA. [3]Bioinformatics Core Unit (BCU), Max Planck Institute for Heart and Lung Research, 61231 Bad Nauheim, Germany. [4]Department of Surgery, University of Maryland School of Medicine, Baltimore, MD 21201, USA. [5]Department of Pediatrics, Herman B Wells Center for Pediatric Research, Indiana University School of Medicine, Indianapolis, IN 46201, USA. ✉e-mail: deqiang.li@nationwidechildrens.org

downstream target gene expression, which is the canonical TGFß signaling pathway[12]. In addition, the binding of ligands to receptors can also directly activate multiple non-Smad mediated intracellular signaling pathways such as MAP kinase pathways and phosphatidylinositol-3-kinase/AKT pathways[13,14]. These non-Smad (noncanonical) TGFß signaling pathways are equally critical for driving some specific cellular processes (e.g., cell proliferation, cell survival, endothelial/epithelial mesenchymal transition)[13,14]. During cardiovascular development, unsurprisingly, TGFß signaling is implicated in most morphological events including cardiac lineage specification, cardiac looping, valvogenesis, and cardiomyogenesis[15]. The TGFß family includes TGFß1, TGFß2, and TGFß3. Although TGFß1, TGFß2 and TGFß3 are abundantly expressed in the developing endocardium and epicardium, it appears that they do not perform redundant functions: individual knockout results in different phenotypes[15]. For instance, global knockout of either *Tgfß2* or *Tgfß3* result in early perinatal lethality, presenting hypoplastic hearts with thin myocardium[16,17], suggesting that they play critical roles for myocardial growth. In contrast, global knockout of *Tgfß1* lead to partial embryonic lethality with survivors appearing phenotypically normal in first the two weeks of life[18]. Nonetheless, it is poorly understood how the expression of TGFß ligands are regulated in the early developing heart.

Epigenetic modulators such as histone deacetylases (HDACs) are critical for early heart development as key regulators of gene expression[19,20]. There are 18 known human HDACs, which can be subdivided into four classes based on their function and DNA sequence similarity. Class I HDACs include HDAC1, HDAC2, HDAC3, and HDAC8, and are predominantly nuclear and ubiquitously expressed[21]. Although class I HDACs have high degree of homology, global knockout of each causes early embryonic lethality[19], suggesting distinct roles during early heart development. Global deletion of *Hdac3* leads to early embryonic lethality at gastrulation[22]. In the developing heart, lack of *Hdac3* in cardiac progenitors disrupts secondary heart field development[23], lymphovenous valve morphogenesis[24], and CM lineage specification[25]. Recently, we reported that HDAC3 in the developing epicardium plays a critical role in supporting compact myocardial wall expansion[26]. It is unclear whether or how HDAC3 in the developing endocardium impacts myocardial trabeculation.

In the current study, we investigated the role of HDAC3 in the developing endocardium during early heart development.

## Results

### Cardiac endocardial KO of *Hdac3* leads to hypotrabeculation

To study the role of HDAC3 in the cardiac endothelium during early heart development, *Hdac3* was specifically deleted in the endothelial cells using *Tie2-Cre*[27], while its expression in other cells including CMs remains intact (Supplementary Fig. 1). Note: since *Tie2-Cre* is transiently expressed in oocytes[28], the mating scheme was always *Hdac3^{f/f}; Tie2-Cre/+* (♂) and *Hdac3^{f/f}* (♀). By recovering the progenies at a series of embryonic and postnatal stages, we identified that *Hdac3* KO in the developing endothelium resulted in early embryonic lethality starting at E12.5 (Supplementary Table 1). To investigate the potential cause of early embryonic lethality, we performed histological analyses. Interestingly, in *Tie2* endothelial *Hdac3* KO embryos (*Hdac3^{tko}*), there was no noticeable phenotype in the various endothelial cell derived tissue where the *Tie2*-lineage contributes, such as endocardium (Fig. 1, Supplementary Fig. 2) and blood vessels (Supplementary Fig. 3), given that *Tie2-Cre* is active in all endothelial cells in the body[29]. In contrast, there were myocardial defects in *Hdac3^{tko}* embryos starting at E11.5: *Hdac3^{tko}* trabeculae appeared to be significantly sparser and shorter as compared to the thick and long webbing like trabecular network in littermate control hearts (Supplementary Fig. 2). This cardiac phenotype in *Hdac3^{tko}* mice was more prominent at E12.5 (Fig. 1a). The compact layer was also significantly thinner in *Hdac3^{tko}* hearts. To further validate the myocardial trabecular phenotype in endothelial *Hdac3* deleted hearts,

we specifically deleted *Hdac3* in the endocardium using *Nfatc1^{Cre/+}*, endocardial specific Cre[30], which also resulted in embryonic lethality (Supplementary Table 2). *Hdac3^{nko}* (*Hdac3^{f/f}; Nfatc1^{Cre/+}*) hearts presented a similar cardiac phenotype, albeit at a later stage (E14.5): smaller and sparser trabeculae, thinner compact wall (Supplementary Fig. 4). This is consistent with the achievement of late deletion of *Hdac3* in the endocardium by *Nfatc1^{Cre/+}* (Supplementary Fig. 4). Nonetheless, these results confirm that the cardiac phenotypes resulting from *Tie2* endothelial *Hdac3* deletion are primarily from the endocardium, and not secondary to another endothelium (e.g., vascular). *Hdac3^{tko}* embryos were used for all subsequent mechanistic studies.

### Decreased CM proliferation in endothelial *Hdac3* KO hearts

Next, we investigated whether altered cell proliferation and/or apoptosis contribute to the hypotrabeculation phenotypes in *Hdac3^{tko}* hearts. At E11.5, the percentage of either p-H3+ or BrdU+ CMs is significantly lower in *Hdac3^{tko}* trabecular and compact myocardium as compared to littermate control hearts (Fig. 1b). This finding is consistent with the hypoplastic cardiac phenotypes seen in *Hdac3^{tko}* hearts (Fig. 1a). In contrast, there was no significant difference in cell apoptosis between *Hdac3^{tko}* hearts and littermate control hearts (Supplementary Fig. 5). Decreased CM proliferation of *Hdac3^{tko}* hearts may result from disrupted communication from the endocardium to the trabecular myocardium. We performed single cell RNA-sequencing (scRNA-seq) on E11.5 *Hdac3^{tko}* hearts and littermate control hearts (Fig. 2a). We obtained high-quality scRNA-seq profiles for 23,468 and 25,547 cells from 4 *Hdac3^{tko}* (*Hdac3^{f/f}; Tie2-Cre/+; R26^{eYFP/+}*) and 4 control (3 of *Hdac3^{f/+}; Tie2-Cre/+; R26^{eYFP/+}* and 1 of *Hdac3^{f/+}; Tie2-Cre/+*) heart single cell suspensions, respectively. Unsupervised clustering successfully separated 6 distinct clusters including endocardial cells and CMs in these samples (Fig. 2b and Supplementary Fig. 6). Referring to a recent transcriptome profiling methodology for clustering embryonic CMs[31], we re-clustered CMs into "trabecular" and "compact" clusters based on the overall scores after applying a few known trabecular and compact markers (Trabecular: *Bmp10*, *Nppa* and *Thbs4*; Compact: *Hey2*, *Mycn* and *Tnnt1*) (Supplementary Fig. 7A and Fig. 2c). Among them, *Bmp10* + CMs nicely represent the trabecular cluster (Fig. 2c), and thus we focused on this population for subsequent analyses since *Hdac3^{tko}* hearts mainly display hypotrabeculation phenotypes (Fig. 1). Overall, the total number of *Bmp10*+ trabecular CMs were not significantly different between *Hdac3^{tko}* and littermate control hearts (Supplementary Fig. 7B). Note: 10,000 cells/sample were sequenced. Next, we performed differential gene expression (DGE) analyses on *Hdac3^{tko}* and littermate control trabecular clusters. We found that the expression of eukaryotic translation elongation factor 1 alpha 2 (*Eef1a2*) and early growth response 1 (*Egr1*) are significantly down regulated in *Hdac3^{tko}* trabeculae as compared to control trabeculae (Fig. 2d). *Eef1a2* and *Egr1* are known to be critical for cell growth[32-34]. These findings are consistent with reduced proliferation index seen in *Hdac3^{tko}* hearts (Fig. 1b), suggesting myocardial growth retardation account for the hypotrabeculation phenotypes in *Hdac3^{tko}* hearts.

### Cardiac endothelial HDAC3 induces ECM gene expression

Among all scRNA-seq samples, 7 of them carried *R26^{eYFP/+}* reporters (3 CTL and 4 *Hdac3^{tko}*). This enabled us to perform a lineage specific analysis. In the single cell suspension, 8.6% cells in CTL hearts and 11.2% cells in *Hdac3^{tko}* hearts were YFP+ (Supplementary Fig. 8A). As expected, the expression of *Hdac3* in YFP+ populations in *Hdac3^{tko}* hearts was significantly decreased compared to YFP+ populations in CTL hearts (Supplementary Fig. 8B). During early heart development, endothelial progenitors can also give rise to mesenchymal cells through endothelial to mesenchymal transition[6,35]. As expected, YFP+ cells (*Tie2* lineage) were primarily CECs with a minority of mesenchymal cells (Fig. 2e). Also, as expected, *Hdac3* expression was absent in *Hdac3^{tko}*

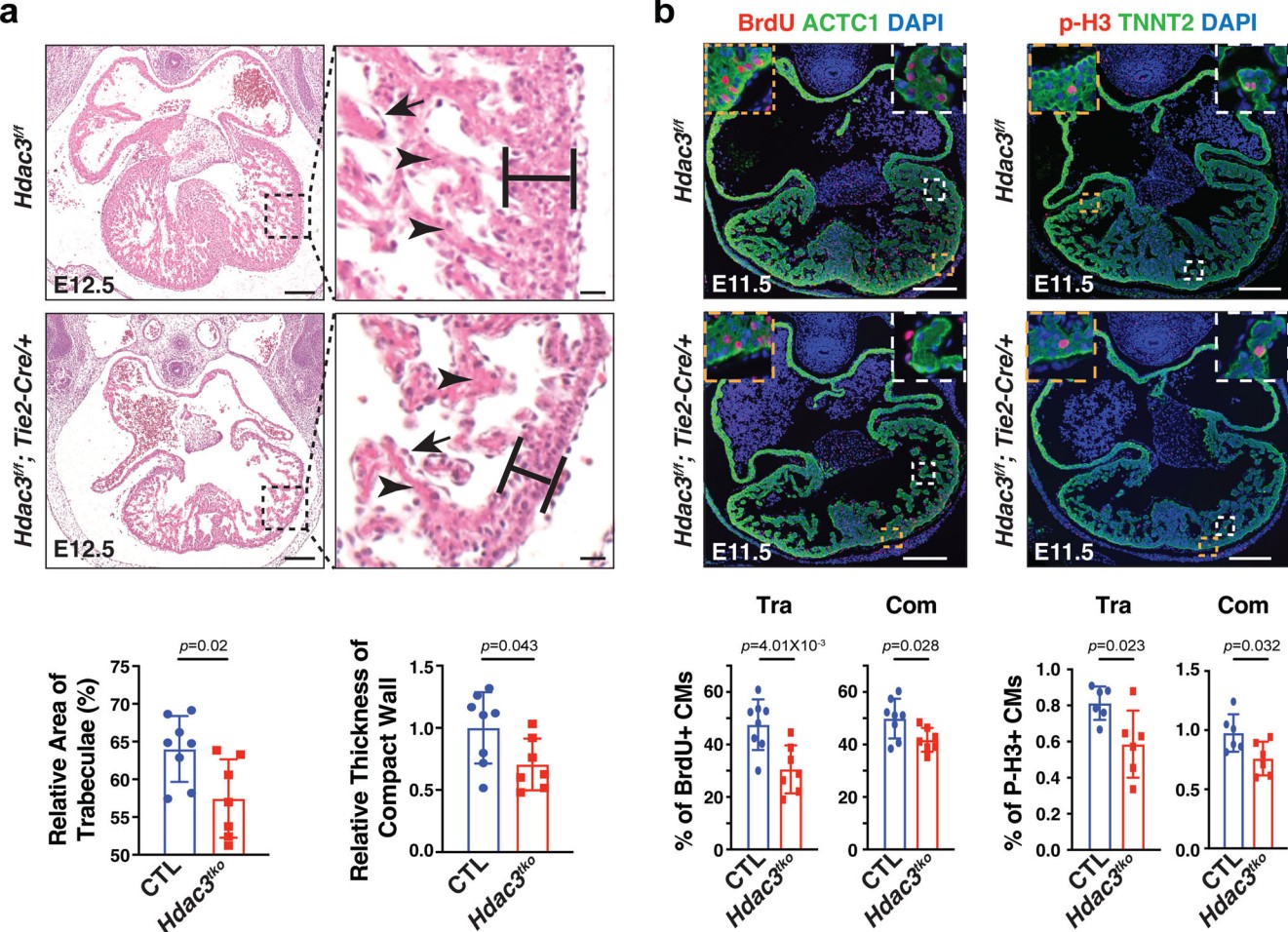

**Fig. 1 | Endothelial specific deletion of Hdac3 results in hypotrabeculation.**
**a** Hematoxylin and eosin staining on cross sections of an E12.5 *Hdac3^tko^* embryo (*Hdac3^f/f^; Tie2-Cre/+*) and a littermate control (CTL) embryo (*Hdac3^f/f^*). Arrows mark endocardium and arrowheads point to trabeculae. Quantifications are shown on the bottom (CTL: $n = 8$, *Hdac3^tko^*: $n = 7$). Scale bars, 200 μm (main panels) and 25 μm (insets). **b** Decreased cardiomyocyte (CM) proliferation in E11.5 *Hdac3^tko^* hearts. Immunofluorescence staining of Bromodeoxyuridine/5-bromo-2′-deoxyuridine

(BrdU) and phospho-Histone H3 (p-H3) staining. High-power Images in the trabecular myocardium (Tra) and in the compact myocardium (Com) are highlighted in white and yellow boxes respectively. Scale bars, 200 μm. Quantification of proliferation index in the trabecular and compact myocardium respectively are shown on the bottom. BrdU (CTL: $n = 8$, *Hdac3^tko^*: $n = 7$); p-H3 ($n = 6$ in each group). Data are presented as the mean ± SD. *P*-values were determined by unpaired two tailed Student's *t* test. Source data are provided as a Source Data file.

YFP+ cells. Subsequent DGE analysis did not reveal significant gene expression differences between *Hdac3^tko^* and control mesenchymal cells (Fig. 2e). Instead, in CEC clusters, we found that the expression of genes regulating cellular component organization was highly enriched in the control YFP + CEC cluster (Fig. 2f). In particular, the expression of ECM associated genes, including *Col13a1, Col1a2, Tgfbi, Timp2, Col26a1, Col3a1, Col4a5, Timp3*, and *P4ha3*, was significantly reduced in *Hdac3^tko^* YFP + CEC cluster as compared to control YFP + CEC cluster (Fig. 2g). *Col1a1, Col1a2, Col2a1*, and *Col3a1* and *P4ha3* DGE was further validated in *Hdac3^tko^* and control heart lysates by qRT-PCR (Fig. 2h). These results suggest that cardiac endothelial HDAC3 induces the expression of ECM associated genes, which are critical for CM proliferation and trabeculation growth (Fig. 2i).

**Endothelial HDAC3 drives CM proliferation through *Tgfβ3***
Trabeculation and ECM remodeling are coordinated during early heart development, and both processes are regulated by growth signaling[36]. To better identify potential growth signaling pathways regulated by HDAC3 in the endocardium, we used a mouse cardiac endothelial cell (MCEC) line[37]. Using CRISPR-Cas9 gene editing technology and the same set of gRNAs we have recently used[26], we successfully knocked out *Hdac3* in the MCECs (Fig. 3a). *Hdac3* KO MCECs grew slower than

*Hdac3* empty vector (EV) control MCECs. To determine whether the secretome from *Hdac3* KO endothelial cells is competent to induce CM proliferation, we treated cultured primary embryonic CMs isolated from E13.5 *Tnnt2^nGFP/+^*[38] hearts with supernatant from either *Hdac3* KO MCECs or EV MCECs. *Hdac3* KO supernatant treatment resulted in significant decrease of the percentage of Ki67+ CMs and the total number of CMs, as compared to EV supernatant treatment (Fig. 3b). These results suggest that growth signals derived from CECs required to support CM proliferation may be inadequate/deficient in *Hdac3* KO MCECs, and the incompetent secretome from *Hdac3* KO endothelial cells may contribute to the hypotrabeculation phenotype. Indeed, treatment of wildtype MCEC supernatants but not control media (DMEM/F-12 without any serum or growth factor supplementation) successfully rescued the hypotrabeculation phenotype in ex vivo cultured E11.5 *Hdac3^tko^* hearts (Supplementary Fig. 11). To unbiasedly search for such growth signals, we performed bulk RNA-Seq analyses on *Hdac3* KO and EV MCECs; we identified 484 downregulated and 239 upregulated genes. Gene ontology pathway analyses revealed that a number of endocardium-initiated signaling pathways were significantly downregulated in *Hdac3* KO MCECs (Supplementary Fig. 9A). Through DGE analysis, we found that growth factors including *Igf1, Tgfβ3, Tgfβ2, Tgfβ1, Igf1, Cxcl12, Nrg1* and *Vegfβ* were significantly

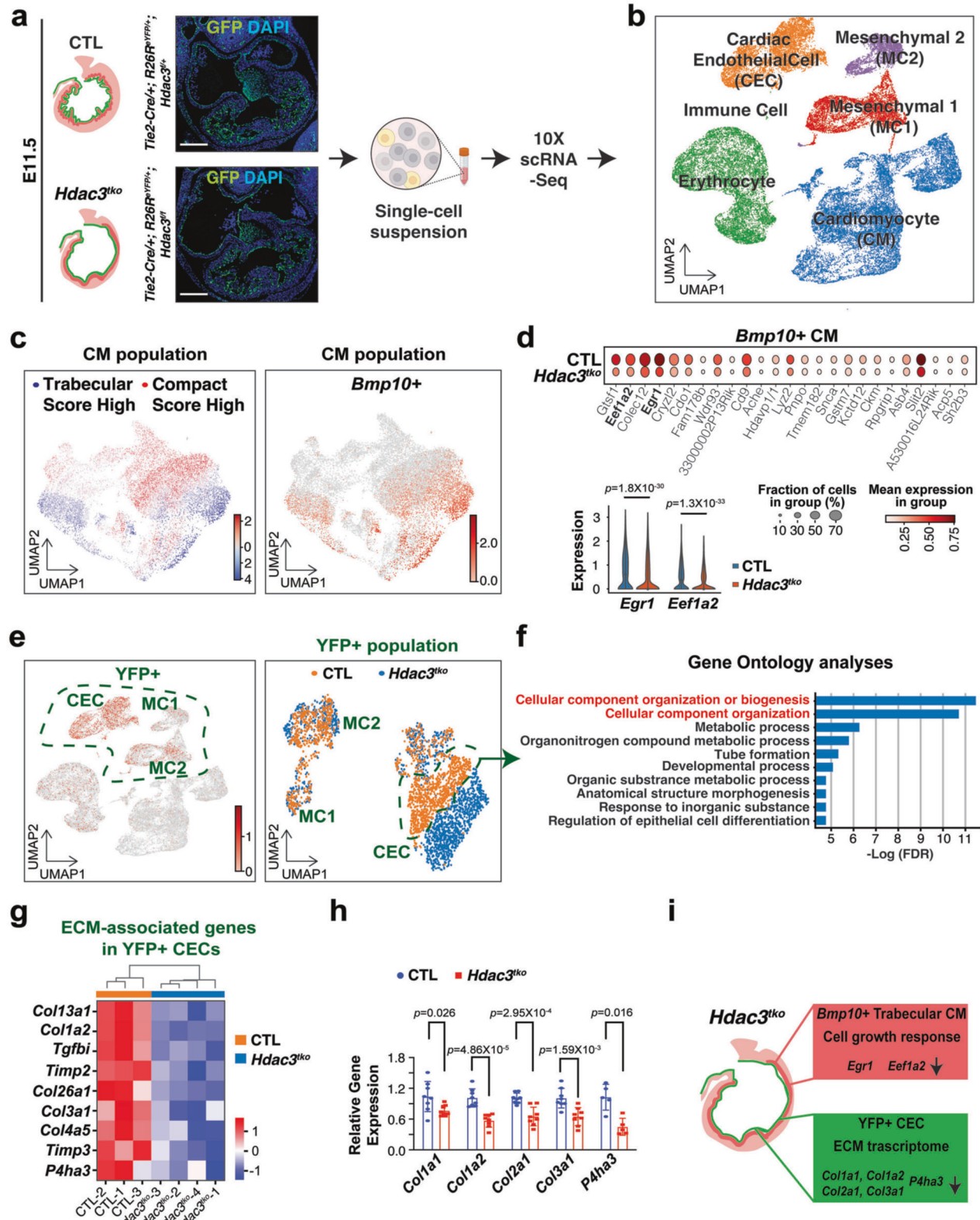

downregulated in *Hdac3* KO MCECs (Supplementary Fig. 9B). We attempted to validate these results at the protein level by western blot. Surprisingly, we found that only TGFβ3 protein expression was significantly decreased in *Hdac3* KO MCECs as compared to *Hdac3* EV MCECs (Supplementary Fig. 9C), and thus we chose TGFβ3 to pursue further. Interestingly, TGFβ3 was also significantly decreased when the deacetylase activity of HDAC3 was inhibited by RGFP966 (a

HDAC3 selective inhibitor) treatment (Fig. 3c and Supplementary Fig. 9D). To assess *Tgfβ3* expression in E11.5 CECs, we sorted Tie2-derived population from *Hdac3*<sup>tko</sup> (*Hdac3*<sup>f/f</sup>; *Tie2-Cre/+; R26*<sup>eYFP/+</sup>) and CTL (*Hdac3*<sup>f/+</sup>; *Tie2-Cre/+; R26*<sup>eYFP/+</sup>) hearts by performing fluorescence activated cell sorting on YFP signal (Supplementary Fig. 10). We found that *Tgfβ3* mRNA level was also significantly downregulated in E11.5 *Hdac3* KO CECs compared to littermate CTL CECs (Fig. 3d). TGFβ3

**Fig. 2 | Single-cell RNA sequencing reveals decreased expression of ECM genes in *Hdac3$^{tko}$* hearts. a** Schematic workflow of scRNA-seq experimental design. Representative micrographs of GFP immunofluorescence staining of E11.5 *Hdac3$^{tko}$* embryo (*Hdac3$^{f/f}$; Tie2-Cre/+; R26R$^{eYFP/+}$*) and a littermate control (CTL) embryo (*Hdac3$^{f/+}$; Tie2-Cre/+; R26R$^{eYFP/+}$*). Scale bars, 250 μm. **b** UMAP visualization of cell populations in an E11.5 control heart. **c** Trabecular and compact cardiomyocyte (CM) score were calculated within *Tnnt2*+ CM population. Left UMAP showed CMs split into a lower cluster that has a high trabecular score, and an upper cluster that has a high compact score. Right UMAP shows the expression level of *Bmp10*+ per cell. **d** Dot plots of differentially expressed genes in *Hdac3$^{tko}$* and CTL trabecular clusters. Violin plots of expression of *Egr1* and *Eef1a2* in *Hdac3$^{tko}$* and CTL trabecular clusters. **e** The UMAP on the left shows that YFP+ cells are distributed in cardiac endothelial cell (CEC, *Cdh*+), mesenchymal cell 1 (MC1, *Pdgfra*+), and mesenchymal cell 2 (MC2, *Dcn*+) clusters. The overlapping UMAPs of *Hdac3$^{tko}$* and CTL YFP+ cells are shown on the right. **f** Gene Ontology pathway analysis of top marker genes in CTL-specific YFP + CEC cluster (Cut off value: -log (FDR) > 5). **g** Heatmap of extracellular matrix (ECM) associated genes in E11.5 *Hdac3$^{tko}$* and CTL hearts. Data were extracted from YFP + CEC clusters in the scRNA-seq dataset. **h** qRT-PCR analysis of *Col1a1, Col1a2, Col2a1, Col3a1* and *P4ha3* in E11.5 *Hdac3$^{tko}$* heart as compared to CTL hearts (n = 8 in each group for *Col1a1, Col1a2, Col2a1* and *Col3a1, n* = 5 in each group for *P4ha3*). Data are presented as the mean ± SD. *P*-values were determined by unpaired two tailed Student's *t* test for **d** and **h**. Source data are provided as a Source Data file. **i** Illustration of decreased expression of proliferation related genes in trabecular CMs and ECM-associated genes in CECs of *Hdac3$^{tko}$* hearts (created with Adobe Illustrator).

downregulation in E11.5 *Hdac3$^{tko}$* hearts was further validated at the protein level by western blot (Fig. 3e). Next, we determined whether TGFβ3 deficiency in *Hdac3* KO MCECs contributed to the CM proliferation deficit following *Hdac3* KO supernatant treatment. First, we quantified TGFβ3 and found that it was significantly decreased in the supernatants from *Hdac3* KO MCECs compared to that from EV MCECs (Fig. 3f). To determine whether TGFβ3 is an essential secretory component of MCEC supernatants in stimulating CM proliferation, we generated *Tgfβ3* KO MCECs using CRISPR-Cas9 gene editing technology (Supplementary Fig. 12A). We found that *Tgfβ3* KO significantly decreased the capacity of MCEC supernatants to induce CM proliferation (percentage of Ki67+GFP+/GFP+ CMs) compared to CTL MCEC supernatant (Supplementary Fig. 12B). Further, we found that *Tgfβ3* KO significantly compromised the capacity of MCEC supernatants in supporting myocardial trabeculation and compaction when treating ex vivo cultured E11.5 CTL hearts with *Tgfβ3* KO MCEC supernatants as compared to control MCEC supernatants (Supplementary Fig. 12C). These results altogether suggest that endocardial TGFβ3 is critical for myocardial growth. Next, we evaluated CM proliferation after we supplemented *Hdac3* KO supernatants with recombinant mouse TGFβ3 protein. Strikingly, TGFβ3 supplementation successfully rescued CM proliferation defects by *Hdac3* KO supernatant treatment (Fig. 3g). Further, we found that TGFβ3 supplementation also rescued myocardial growth deficit in ex vivo cultured E11.5 *Hdac3$^{tko}$* hearts (Supplementary Fig. 13). These results suggest that cardiac endothelial HDAC3 supports myocardial growth by inducing *Tgfβ3* expression.

### HDAC3 induces *Tgfβ3* expression by repressing miR-129-5p
Recently, we reported that HDAC3 regulates growth factors by modulating the miR pathway in the developing epicardium[26]. To identify the potential HDAC3 downstream miR targets in CECs, we performed miR sequencing in *Hdac3* KO and EV MCECs. Through DGE analyses, we found that 5 miRs were significantly upregulated in *Hdac3* KO MCECs (Fig. 4a) (cutoff threshold: Log2 fold change greater than 1.5 and adjusted *P* < 0.01). To identify which miR(s) may regulate *Tgfβ3* expression, we treated MCECs with mimics of each of the 5 miRs and found that treatment with miR-129-5p mimics significantly inhibited the expression of *Tgfβ3* (Fig. 4b). This finding was validated by western blot (Fig. 4c). We further validated the upregulation of miR-129-5p in both *Hdac3* KO MCECs and E11.5 *Hdac3$^{tko}$* hearts (Fig. 4d, e). Interestingly, miR-129-5p was similarly significantly upregulated when the deacetylase activity of HDAC3 is inhibited by RGFP966 treatment (Fig. 4d), suggesting that the inhibition of miR-129-5p expression by HDAC3 is deacetylase dependent. To determine whether miR-129-5p upregulation in MCECs has potential impact on myocardial growth, we treated cultured *Tnnt2$^{nGFP/+}$*[38] CMs or cultured E11.5 wildtype hearts with supernatants from MCECs transfected with either miR-129-5p mimics or scramble RNAs. We found that miR-129-5p supernatant treatment significantly decreased its capacity to induce CM proliferation and myocardial growth (Supplementary Fig. 14). Lastly, to

determine whether HDAC3 induces *Tgfβ3* expression through repressing miR-129-5p, we knocked down miR-129-5p by miRZip lentivirus in *Hdac3* KO MCECs. Remarkably, miR-129-5p knockdown significantly restored the TGFβ3 expression in *Hdac3* KO MCECs (Fig. 4g). Further, the supernatants from miR-129-5p knockdown *Hdac3* KO MCECs significantly restored the CM proliferation promoting capacity from *Hdac3* KO MCEC supernatants (Fig. 4h). These results suggest that HDAC3 promotes the expression of TGFβ3 through repressing miR-129 in supporting trabecular myocardial growth (Fig. 4i).

## Discussion
Myocardial growth/expansion, one of the most critical cardiac developmental events, is accompanied by two separate but closely associated processes: compaction and trabeculation. Many forms of congenital heart disease are associated with abnormal cardiac trabeculation and compaction[2,3]. For proper myocardial development, besides CMs, the surrounding non-CMs including endocardial cells can influence myocardial growth by sending growth signals to the neighboring myocardia[39], as well as secreting ECM proteins that serve as signal transmission media[2,40].

Epigenetic regulation can define a tissue-specific and developmental stage-specific transcription network during early heart development[41,42], mainly achieved by posttranslational modifications of histones, around which DNA winds. There are several different types of histone modifications (e.g., methylation/demethylation, acetylation/deacetylation, phosphorylation/dephosphorylation, ubiquitination/ deubiquitination). By alternating between any of these states, the associated genes can be dynamically poised to be transcriptionally active/repressed. Histone deacetylases (HDACs) are a family of enzymes that remove an acetyl group from histone lysine residues, allowing the histones to wrap the DNA more tightly to repress gene expression. Class I HDACs, which includes HDAC3, are widely expressed, but play specific roles during early embryogenesis and organogenesis[19]. Mesodermal or global KO of *Hdac3* results in myogenic defects and early embryonic lethality[22,23]. Interestingly, specific deletion of *Hdac3* in the myocardium does not generate cardiac morphogenic phenotypes during early heart development, but rather compromises cardiac function at later postnatal stages[23,43]. These early findings suggest that the function of HDAC3 in nonmyocyte compartments may be critical for early cardiac morphogenesis and myocardial development. Our recent work demonstrated that epicardial HDAC3 is critical for driving myocardial compact wall expansion[26]. Our current study suggests that endocardial HDAC3 is important for myocardial trabeculation: endothelial/endocardial deletion of *Hdac3* resulted in hypotrabeculation (Fig. 1 and Supplementary Fig. 4). We note that *Tie2-Cre* turns on around E8.0-8.5 in all endothelial cells[29], whereas *Nfatc1$^{Cre/+}$* is active around E9.5 only in endocardial cells[30] and only achieves efficient *Hdac3* deletion by E14.5 likely due to inefficient IRES-mediated Cre transcription[44] (Supplementary Fig. 4). Nonetheless, these two deletion models generated similar cardiac phenotypes of hypotrabeculation, although we did observe differences in the

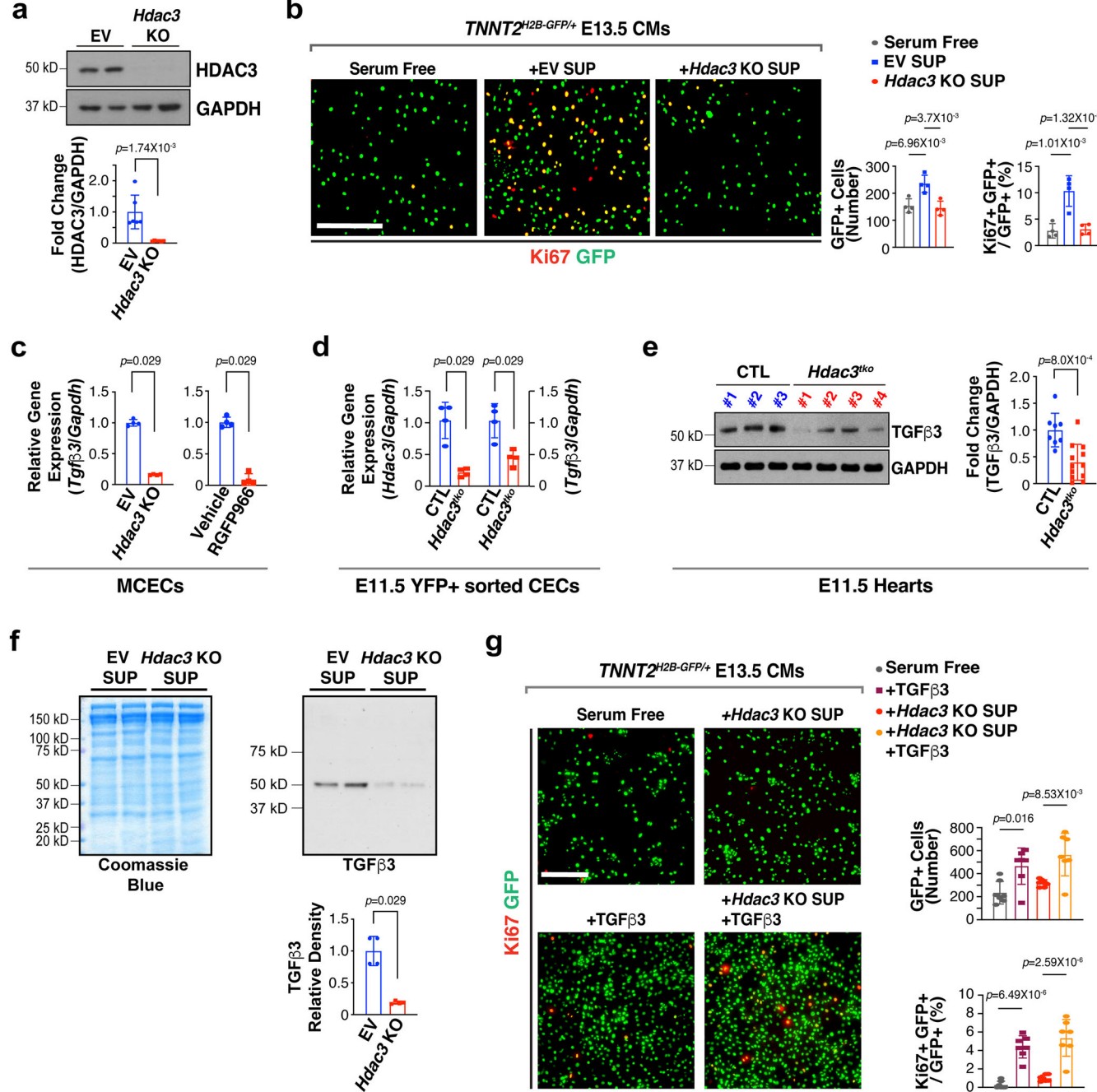

**Fig. 3 | Cardiac endothelial Hdac3 knockout results in downregulation of TGFβ3 and decreased cardiomyocyte proliferation. a** Generation of *Hdac3* knockout (KO) and empty vector control (EV) mouse cardiac endothelial cells (MCECs) by CRISPR/Cas9. Deletion of *Hdac3* was verified by western blot. Quantification is shown below (*n* = 6 in each group). **b** The effects of *Hdac3* KO MCEC supernatants on E13.5 *Tnnt2*^nGFP/+^ cardiomyocyte (CM) proliferation. Representative immunofluorescence micrographs are shown. Scale bar, 275 µm. Percentage of Ki67+ CMs and total number of GFP+ CMs were quantitated (*n* = 4 in each group). **c** Quantification of *Tgfβ3* expression in *Hdac3* KO MCECs, or RGFP966-treated MCECs (10 µM) by qRT-PCR. *Gapdh* was used as cDNA loading control (*n* = 4 in each group). **d** Downregulation of *Tgfβ3* gene expression in *Hdac3* deficient cardiac endothelial cells. YFP + E11.5 cardiac endothelial cells (CECs) from *Hdac3*^tko^ (*Hdac3*^f/f^; *Tie2-Cre/+; R26*^eYFP/+^) and CTL (*Hdac3*^f/+^; *Tie2-Cre/+; R26*^eYFP/+^) hearts were sorted by fluorescence-activated cell sorting. Quantification of *Hdac3* and *Tgfβ3* expression by qPCR was shown on the right. *n* = 4 for each group. *Gapdh* was used as cDNA

loading control (*n* = 4 in each group). **e** Representative western blot and quantification of TGFβ3 expression in E11.5 *Hdac3*^tko^ hearts (*n* = 12) and littermate CTL hearts (*n* = 8). **f** Secretion of TGFβ3 from *Hdac3* KO and EV MCECs. Coomassie brilliant blue staining of total extracted proteins from supernatants served as protein loading controls. TGFβ3 in the MCEC supernatants were detected by western blot. Quantifications are shown on the right (*n* = 4 in each group). **g** Rescue of CM proliferation defect by *Hdac3* KO MCECs supernatant supplemented with recombinant TGFβ3 (working concentration: 250 ng/ml). Representative immunofluorescence micrographs are shown. Scale bar, 200 µm. Percentage of Ki67+ CMs and total number of GFP+ CMs were quantitated (*n* = 7 in each group). Data are presented as the mean ± SD. *P*-values were determined by unpaired two tailed Student's *t* test for **a** and **e**, two tailed Mann–Whitney U test for **c**, **d** and **f** and one-way ANOVA followed by the Tukey post hoc test for **b** and **g**. Source data are provided as a Source Data file.

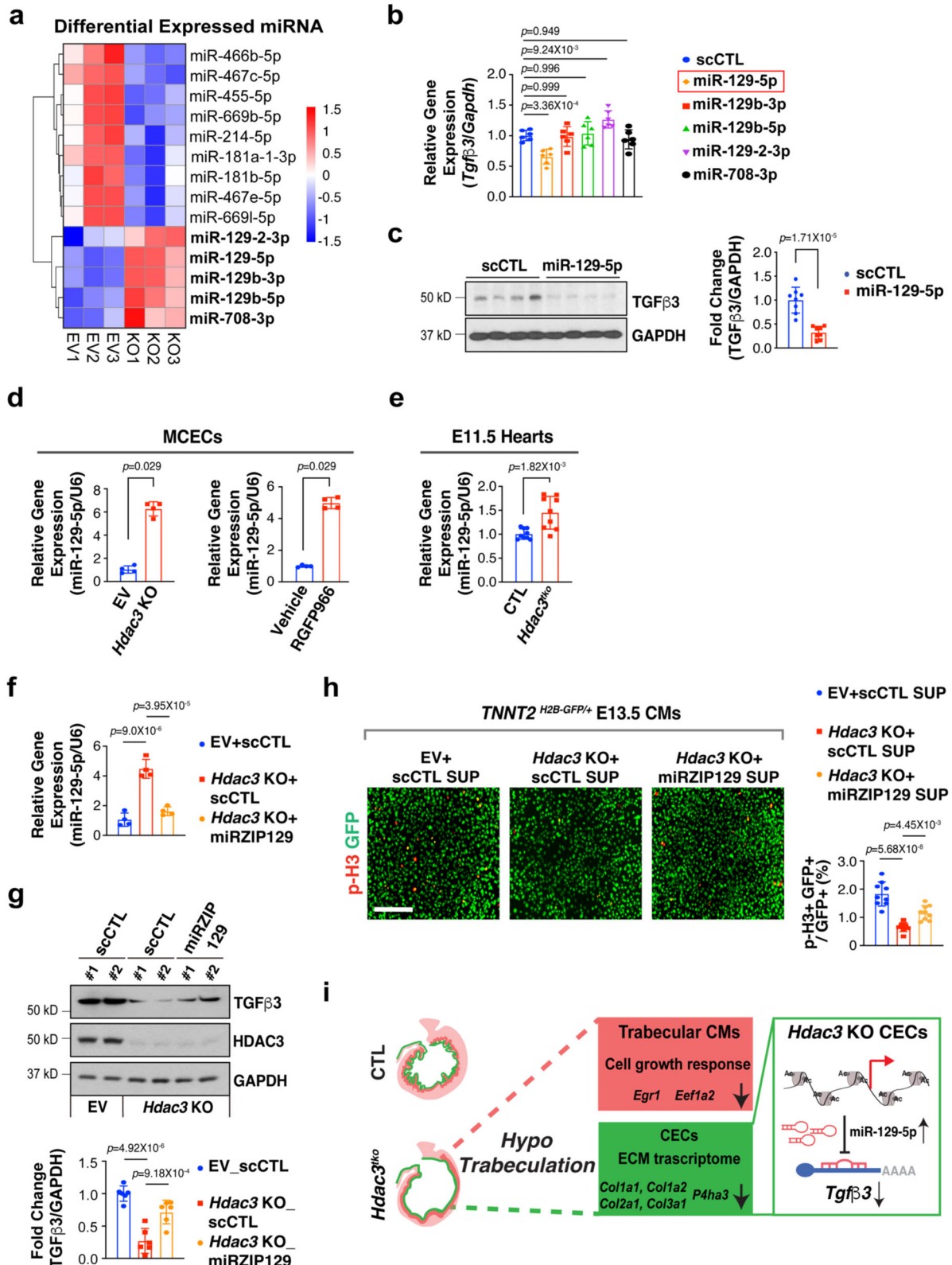

severity of the myocardial phenotypes and time of lethality, which we attribute to the different temporal and spatial activities and/or efficiencies of these two Cres. We also observed that both E12.5 $Hdac3^{tko}$ and E14.5 $Hdac3^{nko}$ hearts exhibited thin compact wall phenotype. This is likely attributable to $Hdac3$ deficiency in the subepicardial endothelial population which form coronary arteries invading and supporting compact wall[45]. This late thin compact wall phenotype is also

consistent with the similar phenotype resulted from epicardial deletion of $Hdac3$[26]. An earlier report of conditional deletion of $Hdac3$ in endothelial cells using a different $Tie2$-$Cre$ driver primarily targeting hemogenic endothelial cells[27] resulted in mid-gestational lethality (E14.5) presenting predominantly a lymphovenous and lymphatic valve malformation phenotype[46]. However, it is unknown what phenotypes were present in the myocardium in that study as none were

**Fig. 4 | HDAC3 induces Tgfß3 by repressing miR-129-5p. a** Heatmap of microRNA (miR) sequencing of *Hdac3* knockout (KO) and empty vector control (EV) mouse cardiac endothelial cells (MCECs). MiRs with reads less than 100 were discarded and miRNA expression levels were normalized by TPM (transcript per million) values (TPM = [miRNA total reads/total clean reads] × 10⁶). Log₂ fold changes were calculated by TPM per miR in the KO group divided by the mean TPM per miR in the EV group then followed by calculation of Log₂. Significantly downregulated miRs are shown in blue, and significantly up-regulated miRs are shown in red. *P*-values were determined by DeSeq2 (One sided). Cut-off criteria: Log₂ > 1.5, adjusted *P*-value < 0.01. **b** Quantification of the expression of *Tgfß3* in MCECs after miR mimics treatment (final concentration 10 nM) by qRT-PCR. *Gapdh* was used as a cDNA loading control (*n* = 6 in each group). Relative gene expression was compared to the miR scrambles (scCTL) group. **c** Representative western blot and quantification of the expression of TGFβ3 after miR-129-5p mimics treatment (*n* = 8 in each group). **d** (Left) Quantification of *Tgfß3* expression in *Hdac3* KO and EV MCECs (*n* = 4 in each group). (Right) Quantification of the expression of *Tgfß3* in MCECs after Hdac3 inhibitor treatment (RGFP966, final concentration 5 μM) (*n* = 4 in each group). **e** Quantification of the expression of *Tgfß3* in *Hdac3^tko^* heart lysates (*n* = 9 in each group). **f** Quantification of the expression of miR-129-5p by qRT-PCR. *Hdac3*

KO or EV MECs were transfected with LentimiRa-GFP-miRZip129 (miRZIP129) or pGreenPuro Scramble Hairpin control lentivirus (scCTL) and transfected cells were selected using puromycin (*n* = 4 in each group). **g** Representative western blot and quantification of the expression of TGFβ3 after treatment with miRZIP129 or scCTL in *Hdac3* KO and EV MCECs (*n* = 6 in each group). **h** Knockdown of miR-129-5p rescues CM proliferation deficit induced by *Hdac3* KO MCEC supernatants. Representative immunofluorescence micrographs are shown. Scale bar, 200 μm. Quantification of percentage of phospho-histone H3 (p-H3)+ CMs was shown on the right. Independent samples: *n* = 9 in each group. Data are presented as the mean ± SD. *P*-values were determined by unpaired two tailed Student's *t* test for **c** and **e**, two tailed Mann–Whitney U test for **d** and the one-way ANOVA followed by Tukey post hoc test for **b**, **f**, **g** and **h**. Source data are provided as a Source Data file. **i** Working model (created with Adobe Illustrator): in the developing endocardium, HDAC3 represses the expression of miR-129-5p to release its suppression on the expression of TGFβ3. When *Hdac3* is deleted, the expression of miR-129-5p is increased, which subsequently suppresses the expression of TGFβ3 to a stronger extent, and the decrease of TGFβ3, together with decreased ECM, leads to hypotrabeculation.

reported. Nonetheless, we deemed it inapplicable to evaluate potential lymphatic valve phenotypes in our pan-endothelial *Hdac3* KO embryos since these mice died around E12.5 (Supplementary Table 1), while lymphovenous and lymphatic valve formation starts at E13.5[47]. However, it is interesting to note that the role of HDAC1 and HDAC2 in the developing endocardium appears to be dispensable, as deletion of either *Hdac1* or *Hdac2* by *Tie2-Cre* do not elicit apparent cardiac phenotypes[24]. These results suggest that HDAC3 plays distinctive roles from HDAC1 and HDAC2, although they share high homology and are grouped in the same HDAC subfamily[19].

TGFß signaling is involved in many processes during early heart development such as ECM production, endothelial to mesenchymal transition, and myogenesis[11,48–50]. Specifically, TGFß signaling plays a critical role during early cardiac myocardial development[51–54]. For instance, knockout of either TGFß2 or TGFß3 results in hypoplastic ventricular myocardium[15], while TGFß1 is reported to inhibit CM proliferation[54,55], suggesting the complexity and specificity of TGFß isoforms during early heart development. In the current study, we found that TGFß3 expression was significantly decreased in *Hdac3* deleted endocardium, and TGFß3 treatment vigorously stimulated CM proliferation (Fig. 3). Interestingly, inhibition of deacetylase activity of HDAC3 alone significantly decreased TGFß3 expression. Further, supplementation of TGFß3 to cardiac endothelial *Hdac3* KO secretome significantly rescued CM proliferation and myocardial growth defect (Fig. 3), suggesting that cardiac endothelial HDAC3 supports trabeculation through inducing TGFß3 in a deacetylase dependent manner. Interestingly, HDAC3 is also reported to inhibit TGFß1 expression in the second heart field development in a deacetylase independent manner[23], whereas HDAC3 deteriorates colorectal cancer progression by inducing TGFß1 through inhibiting miR-296[56]. These divergent findings illustrate that HDAC3 can either promote or inhibit TGFß signaling pathway through different molecular mechanisms in a tissue/cell dependent manner.

Besides functioning as growth signals to CMs, TGFß signaling is predominantly known for its induction on ECMs[49,57]. During early heart development, ECM is mainly secreted from endocardium and mediates signal transduction from endocardium to myocardium supporting trabeculation[56]. Our scRNA-Seq data revealed that ECM is significantly down regulated in *Hdac3* endothelial KO hearts (Fig. 2). P4ha3, a major enzyme for collagen synthesis[58] and a TGFß downstream target[59], was also significantly decreased in *Hdac3* endothelial knockout hearts (Fig. 2). These results suggest that ECM downregulation may be secondary to decreased TGFß signaling to contribute to the hypotrabeculation phenotypes in *Hdac3* endothelial/endocardial knockout hearts.

We identified that HDAC3 suppresses miR-129-5p in inducing TGFß3 (Fig. 4). miR-129-5p plays important roles in cancer by either promoting or inhibiting tumorigenesis through regulating a variety of downstream targets and pathways[60,61]. Several studies have shown that miR-129-5p protects adult hearts from ischemia/reperfusion injuries[62,63], while another study demonstrated that treating cultured H9c2 cells with HDAC pan-inhibitor Trichostatin A results in decreased CM proliferation with enhanced miR-129-5p expression[64]. This is consistent with our current findings in the developing endocardium (Fig. 4). In our recent study, we found that HDAC3 induces the expression of IGF2 and FGF9 by suppressing miR-322/503 in the developing epicardium[26]. Again, these findings altogether suggest that HDAC3 can induce different sets of growth signals by switching their downstream miR targets in a tissue/cell context dependent manner. On the other hand, HDAC3, as a histone modifier, may have many downstream targets that participate in other signaling pathways and exert various other biological effects. For instance, several Wnt ligands (e.g., *Wnt4*, *Wnt5a*, *Wnt11*) and Hedgehog signaling components (e.g., *Smo*, *EVC*, *Gli2*) were also significantly downregulated in *Hdac3* KO MCECs. The disruption of these signaling pathways may affect cell adhesion, migration and ciliogenesis[65,66], thus also contribute to the overall *Hdac3^tko^* cardiac phenotypes. We will be actively investigating these possibilities in our future studies.

In summary, our current study demonstrates that endocardial HDAC3 promotes myocardial trabeculation by inducing TGFß signaling through repressing miR-129-5p during early heart development. This finding, along with others, exemplifies the importance of endocardial signaling for driving myocardial morphogenesis, and thus provides insights on the pathogenesis of congenital heart disease. The dual epigenetic inhibition mechanism may be leveraged in some other contexts such as adult heart repair/regeneration.

## Methods

### Ethics declarations
All animal protocols were approved by the Nationwide Children's Institutional Animal Care and Use Committee (IACUC #AR22-00194).

### Mice
*Hdac3^flox/+^*, *Tie2-Cre*, *Nfatc1^Cre/+^*, *R26^eYFP^* and *Tnnt2^nGFP/+^* mice were previously described[27,30,38,67,68]. B6.Cg-Tg(Tek-cre)12Flv/J (abbreviated as *Tie2-Cre* in this manuscript), *Hdac3^flox/+^*, and *R26^eYFP^* mice are available at the Jackson Laboratory (Stock numbers: #004128, #024119 and #006148, respectively).

## Histology and immunofluorescence staining

All embryo specimens were fixed in 4% paraformaldehyde overnight, dehydrated through an ethanol series, paraffin embedded, and sectioned (6-7μm). Primary antibodies (Supplementary Data 1) were incubated at 4 °C overnight and secondary antibodies (Alexa 488 or Alexa 555, Thermo Fisher Scientific, Waltham, MA) were incubated at room temperature for one hour. Stained slides were imaged on a Zeiss LSM 710 confocal Microscope or a Leica DM6 fluorescence microscope. Epifluorescence and gross images of embryos were imaged on a Leica M205 FCA stereo fluorescence microscope. The number of phosphor-Histone 3 (p-H3)+ cells, Bromodeoxyuridine (BrdU)+ cells, and TUNEL+ cells was quantified using ImageJ software.

## Embryonic cardiomyocyte (CM) culture and proliferation assessment

E13.5 ventricular CMs were isolated from dissected embryonic hearts and cultured as previously described[69]. To assess cell proliferation, CMs were fixed with 4% paraformaldehyde and stained with Ki67 or p-H3 antibody followed by Alexa 594-conjugated secondary antibody (Thermo Fisher Scientific, Waltham, MA). The immunostaining was visualized and imaged on a Keyence BZ-X800 microscope. The percentages of Ki67+GFP+/GFP+, p-H3+GFP+/GFP+ and total number of GFP+ CMs were quantified using ImageJ software.

## Endothelial cell culture, transient transfection, and lentiviral infection

Mouse cardiac endothelial cells (MCECs) were purchased from Cedarlane Labs (Catalogue #CLU510, Burlington, NC). Cells were tested for mycoplasma contamination and resulted negative. MCECs were cultured in 10% fetal bovine serum (FBS) supplemented DMEM at 37 °C in a humidified incubator with 5% $CO_2$. To generate stable Hdac3 KO MCECs, as previously described[26], we cloned Hdac3 sgRNA into lenti-CRISPR v2 vector (Addgene #52961; Watertown, MA). Seventy-percent confluent Lenti-X™ 293T cells (Takara Bio USA, Inc., Mountain View, CA) were transfected with Hdac3 lentiviral plasmids (Hdac3 KO or empty vector control [EV])[26], Tgfβ3 lentiviral plasmids (Tgfβ3 KO or empty vector control [EV]) were purchased from VectorBuilder, PsPAX2 (Addgene #12260) and PMD2G (Addgene #12259). Seventy-two hours after transfection, supernatants containing lentivirus were collected and filtered through a 40μm cell strainer. Mir-129-5p, miR-129b-3p, miR-129b-5p, mir-129-2-3p, miR-708-3p mimics and scramble control mimics were custom-ordered from ThermoFischer Scientific (Waltham, MA). MCECs were transfected with miRNA mimics or control mimics (30 pmol) using Lipofectamine RNAiMAX (Thermo Fisher Scientific, Waltham, MA). 72 h after miR mimics treatment, supernatants were collected. miRZip anti-miR precursor constructs for miRZip-129-5p and pGreenPuro Scramble Hairpin control constructs were purchased from System Biosciences (Palo Alto, CA), and lentiviruses were generated according to the manufacturer's protocol.

## Ex Vivo embryonic heart culture

E11.5 embryonic hearts were dissected in cold PBS, and then cultured in DMEM/F-12 media (Thermo Fisher Scientific, Waltham, MA, #11039021). Embryonic cardiac explants were treated with either TGFß3 recombinant protein (Sigmal Aldrich, Saint Louis, MO, #SRP6552, final concentration: 250 ng/ml), Tgfβ3 KO MCEC supernatants, miR-129-5p mimics-treated MCEC supernatants and their corresponding control media or control MCEC supernatants for 24 h at 37 °C. Hearts were then fixed in 4% paraformaldehyde overnight at 4 °C and processed for paraffin embedding and sectioning.

## Fluorescence activated cell sorting of YFP+ Tie2 derivative cardiac cells

E11.5 hearts were dissected and dissociated with 0.5% trypsin−EDTA into a single cell suspension. Cell suspension was washed with PBS and YFP+ cells were gated using their autofluoresence signal and sorted and using a BD Influx cell sorter (BD Biosciences, Franklin Lakes, NJ). The harvested YFP+ cells were used for subsequent gene expression analysis.

## RNA isolation, quantitative real-time PCR (qRT-PCR), and bulk RNA-Seq

MCECs and embryonic day (E) 11.5 hearts were used for RNA isolation. E11.5 hearts were microdissected in cold PBS and snap frozen in liquid nitrogen. The RNeasy Plus Mini Kit (QIAGEN, Germantown, MD) was used to extract total RNA, and cDNAs were generated using the Superscript III kit (Thermo Fisher Scientific, Waltham, MA). To detect mRNA, SYBR Green qRT-PCR was performed. PCR primers for genes of interest are listed in Supplementary Data 1. To perform mature microRNA expression analyses, purified total RNAs were converted to cDNAs using TaqMan™ MicroRNA Reverse Transcription Kit (Catalogue #4366596, Thermo Fisher Scientific, Waltham, MA). qRT-PCR reactions were run on StepOne Plus Real-Time PCR System (Applied Biosystems, Waltham, MA). The probe sequences for mmu-miR-129-5p and U6 snRNA were purchased from Thermo Fisher Scientific. For bulk RNA and miRNA-Sequencing, samples were prepared following the provider's guidelines (Novogene Corporation Inc, Sacramento, CA) and sequenced on an Illumina NextSeq500 (San Diego, CA). Sequencing reads were aligned to the UCSC mm10 reference genome using tophat2 and bowtie2 in R. Differential expression of transcripts was calculated using the cufflinks suite in R analyses.

## Single cell RNA-Sequencing (scRNA-seq)

E11.5 heart tissues were dissected in cold PBS and digested with 0.25% trypsin at 37 °C for 15 min. Digested cardiac cells were filtered with 70 μm cell strainers to remove remaining clumps. Subsequently, 10,000 live cells were captured with Chromium Next GEM Single Cell 3′ Reagent Kits v3.1 (10X Genomics, Pleasanton, CA) following manufacturer's instructions. Briefly, single cells were partitioned into nanoliter-scale Gel Beads-in-emulsion (GEMs) using Chromium Next GEM Chip G in the Chromium Controller. Immediately following GEM generation, the Gel Bead was dissolved, primers were released, and co-partitioned cell was lysed, and its mRNA were reverse transcribed into barcoded cDNA. After further cleanup and amplification, the cDNA was enzymatically fragmented and amplified via PCR to generate sufficient mass for library construction. After end repair, A-tailing, adapter ligation and sample index PCR, the library comprised constructs containing the sample index, UMI sequences, barcode sequences, and Illumina standard sequencing primers P5 and P7 at both ends. The library was sequenced with Illumina NextSeq S4 platform to a sequencing depth of 140-270 million reads per sample. Reads were aligned to the Ensembl 104 mouse reference genome including the sequence for eYFP as a separate contig using STARsolo (version 2.7.10b)[70]. The resulting count tables (cells × genes) were converted into Annotated Data (anndata) format for analysis in Python. For quality control, all cells with less than 1500 or more than 9000 genes expressed were excluded. Additionally, cell doublets were filtered out using the Scrublet method[71]. Gene counts were normalized per cell to a total target sum and transformed using log1p. Highly variable genes were identified using the Seurat method[72] and these genes were utilized for dimension reduction with PCA. Finally, cell embedding was performed with the UMAP algorithm (umap-learn version 0.5.3) and cells were clustered using leiden (leidenalg version 0.9.1) with resolution 0.2. Scoring of trabecular and compact CM subtypes was performed using 'score_genes' and differentially expressed genes between groups were obtained using 'rank_genes_groups', both methods from the SCANPY package[73].

## Western blotting

Cell or tissue lysates were prepared in lysis buffer (20 mM Tris-HCl [pH 7.5], 15 mM NaCl, 1 mM $Na_2EDTA$, 1 mM EGTA, 1% Triton X-100, 1 μg/ml

leupeptin, 2.5 mM sodium pyrophosphate, 1 mM $Na_3VO_4$, and 1 mM β-glycerophosphate) with protease inhibitor cocktail (Roche, Indianapolis, IN) and 1 mM phenylmethylsulfonyl fluoride. Protein samples were resolved on 4–12% SDS-PAGE acrylamide gels before transferring to polyvinylidene fluoride membranes. Primary antibodies were visualized by chemiluminescence using HRP-conjugated secondary antibodies. Antibodies are listed in Supplementary Data 1. Western blots were normalized to loading controls and densitometric analysis was performed using Image J software.

## Statistical analysis and reproducibility

All experiments were independently repeated at least three times, and the number of samples (n) is stated in figure legends. Results are reported as the mean ± SD. For statistical analyses, normal distribution of data was assessed by Shapiro-Wilk's test (when $n \geq 6$). For data with small sample size ($n < 6$), non-parametric tests were applied. For parametric data with normal distribution, the statistical significance of the difference between means was assessed using the unpaired two tailed Student's t-test between two groups and the Tukey or Dunnett post hoc test for one-way ANOVA. For non-parametric data or data with small sample size ($n < 6$), the Mann–Whitney U test and Kruskal–Wallis test for one-way ANOVA followed by the Dunn post hoc test were performed for comparisons between two groups. Specific statistical tests are stated in figure legends. RNA-seq data were background-corrected, variance-stabilized, normalized, and count matrix extracted through quality control using lumi package for the R programming language (http://bioconductor.org/packages/release/bioc/html/lumi.html). Differential expression analysis was performed using DESeq2 R package (https://bioconductor.org/packages/release/bioc/html/DESeq2. html). The resulting p-values were adjusted using the Benjamini and Hochberg's approach for controlling the False Discovery Rate (FDR)[74]. Genes or miRs with an adjusted p-value < 0.01 found by DESeq2 were assigned as differentially expressed. Gene ontology (GO) enrichment analyses were performed using the Database for Annotation, Visualization and Integrated Discovery (DAVID) bioinformatics resources (https://david.ncifcrf.gov/). For relative gene expression analysis of qRT-PCR data, the expression of a gene of interest relative to an internal loading control gene was calculated using the comparative CT method (-ΔΔCT)[75]. The relative protein expression on western blots was quantified by densitometry, Fold changes for relative gene or protein expression were derived through division of values in the experimental group by the mean values from the control group. P-value < 0.05 was considered statistically significant. All statistical analyses were performed using GraphPad Prism 8 (GraphPad Software, Boston, MA). The images with the value closest to the mean value were selected as representative images. We state the number of biological samples to generate the data shown in this manuscript. No data were excluded from the analyses.

## Reporting summary

Further information on research design is available in the Nature Portfolio Reporting Summary linked to this article.

## Data availability

All data associated with this study are present in the main text or the supplementary materials. Source data are provided with this paper. The scRNA-seq, RNA-seq and miRNA-seq data generated in this study have been deposited in the GEO database under accession code GSE229661. All raw data are available from the authors upon reasonable request. Source data are provided with this paper.

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

## Acknowledgements

We thank Dr. Dennis J. Lewandowski in the Heart Center at the Nationwide Children's Hospital for his critical reading and editing of the manuscript. This work was partially supported by the National Heart, Lung, and Blood Institute R01 grant HL153406 (D.L.), American Heart Association (AHA)-Career Development Award 23CDA1046244 and AHA-Innovative Project Award 23IPA1048560 (J.J.).

## Author contributions

J.J. and D.L. conceived and designed the study, analyzed the experiments, wrote the paper. M.B and M.L performed scRNA-Seq data analysis. Y.J.K and E.K. performed the experiments. C.C. provided *Tnnt2^{nGFP/+}* mice. V.G., J.J. and D.L. edited the manuscript. All authors reviewed the results and approved the final version of the manuscript.

## Competing interests

The authors declare no competing interests.
