## [Peer Review File · Nature Communications]

REVIEWER COMMENTS

Reviewer #1 (Remarks to the Author):

Summary

In this study, the authors investigated the role of HDAC3 in the developing endocardium during early heart development. As normal development progresses, inner cardiac endothelial cells take part into trabeculation. Failure of proper ventricular trabeculation is often associated with congenital heart disease. HDAC3 in the developing epicardium plays a critical role in supporting compact myocardial wall expansion, and it is unclear whether or how HDAC3 in the developing endocardium impacts myocardial trabeculation.

To study the role of HDAC3 in the cardiac endothelium during early heart development, Hdac3 was specifically deleted in the endothelial cells using Tie2-Cre. Hdac3 KO in the developing endothelium resulted in early embryonic lethality starting at E12.5. Histological analysis in embryos at E11.5 showed myocardial defects consisting of Hdac3tko trabeculae appeared to be significantly sparser and shorter as compared to the thick and long webbing like trabecular network in littermate control hearts.

General comments

The article is well written, and data are well presented and explained. The phenotype of knocking out Hdac3 is also clear. In fact, the authors demonstrate that Hdac3 KO results in hypotrabeulation, and culturing E13.5 CMs with media from a Hdac3 KO mouse cardiac endothelial cells (MCECS) line was shown to block CM proliferation. Also, culturing E13.5 CMs with media supplemented with recombinant TGFB3 rescued the proliferation capacity of CMs.

In my opinion, what is probably missing, is the confirmation of the involvement of HDAC3 downstream regulators (miR-129-5p and TGFB3) in producing the Hdac3 KO phenotype. Considering that TGFB3 KO is lethal perinatally, the phenotypic effect of overexpressing miR-129-5p in endothelial cells should produce the same effect. Also, KO of TGFB3 in endothelial cells should give the same hypotrabeulation as Hdac3 KO in the developing heart. I also would expect that overexpression of miR-129-5p to result in hypotrabeulation and possibly lethality as both Hdac3 and TgfB3 KO are lethal. Also, culturing E13.5 CMs with media supplemented with media from MCECs in which miR-129-5p has been silenced should rescue the phenotype of CM proliferation.

Finally, the mouse cardiac endothelial cell (MCECs) helped the authors identifying the mechanism downstream Hdac3 KO. Even though, data are significantly comparable with E12.5 primary cells or hearts, the results and statistical significance observed in control and Hdac3 KO are more striking in MCECs than in primary cells or tissues. This is shown in Figure 3 D and figure 4E and G.

1. Tgfb3 expression reduction in Hdac3 KO MCECs or via HDAC3 inhibitor and in E11.5 is significant. Quite different is the percentage of downregulation in Hdac3 KO and Hdac3 inhibitor between MCECs and in E11.5 hearts (Figure 3 C and D. In Figure 3 D, the KO of Tgfb3 is 10% in E11.5 hearts in comparison to MCECs (80-90% KO). Is 10% enough to cause the phenotype of hypotrabeulation and perinatal lethality? In favour of the authors, supplementation of recombinant TGFB3 in E13.5 CMs rescues proliferation defects due to Hdac3 KO.

2. As a point for positive control, could the authors show the expression levels of HDAC3 in the scRNA-seq data, showing the loss of HDAC3 specifically in the endocardial cell population on the same UMAP for Figure 2E – Especially those marked by Tie+ expression?

3. For Figure 1B, could the authors add a heatmap/clustering profile in supplementary to show specific markers used to separate out the clusters?

4. While the effect on TGFB3 is clear and convincing, is there a secondary independent sgRNA/siRNA strategy used to validate the phenotype of HDAC3 KO in MCECs – To account for possible off targets?

5. It is interesting that TGFB3 can enhance proliferation in E13.5 CMs, it might be work looking at TGFB3 supplementation in adult CMs, and see if there is any effect on proliferation at the later stages.

6. In Figure 3G, the authors suggests that HDAC3 induces TGFB3 expression through repressing miR-129-5p, using miRZip lentivirus. The modest rescue in Figure 3G may suggest that TGFB3 expression may not be entirely regulated by miR-129-5p. Might there be other miRNAs or regulators be synergistically regulating its expression? Or other possible explanations? The authors could discuss this further.

7. Figure 4G. Authors show a modest rescue of miR-129-5p and TGFB3 after miRZIP129 in Hdac3 KO. One possible experiment could be silencing miR-129-5p in MCECs and supplementing E13.5 CMs with supernatant to rescue CM proliferation and strengthen the downstream role of miR-129-5p.

8. TGFB3 supplementation rescue CM proliferation defects due to Hdac3 KO supernatant treatment in E13.5 CMs. Would be good to measure miR-129-5p presence in supernatant. miR-129-5p should be low in presence of TGFB3 supplementation and up in presence of Hdac3 KO.

9. Would be interesting to see a possible effect of miR-129-5p silencing in primary cells in vitro or in vivo. Silencing of miR-129-5p might induce proliferation in primary CMs.

10. Mechanistically, the mechanism downstream HDAC3 is explained. Inhibition or KO of HDAC3 upregulates miR-129-5p which in turn downregulates TgfB3. What's probably missing is the phenotypic effect of the downstream regulator miR-129-5p. Upregulation of miR-129-5p or KO of TGFB3 in endothelial cells should give the same hypotrabeulation as Hdac3 KO in the developing heart. Interestingly, KO of TgfB3 in embryo is lethal as well as Hdac3 KO. It would be interesting to see whether miR-129-5p upregulation is also lethal, considering its function in between Hdac3 and TgfB3.

11. The authors should also define what they mean by "double brake axis" in their title, as there were no reference or explanation to what it means in the main text.

Minor comments

1. Figure 4. Typo at lines 744 745 746. It is the quantification of miR-129-5p and not TgfB3.

Reviewer #2 (Remarks to the Author):

The manuscript by Jang et al reports a new signaling pathway mediated by HDAC3 that is highly relevant to congenital heart disease (CHD).

Epigenetic regulation of mammalian development is well studied. HDAC3 has been linked to embryogenesis and multiple organ development by several studies (Reviewed in detail by Dsilva et al in *Mol Reprod Dev*, 2022, 90(1): 14-26). For example, HDAC3 was shown to physically interact with Tbx5 to regulate Tbx5 acetylation, which represses Tbx5-dependent expression of cardiac lineage-specific genes, supporting a direct role of HDAC3 in early cardiogenesis (Lewandowski et al, 2014, *Human Mol Genet*, 23(14):3801-3809).

The authors previously demonstrated a signaling pathway through which HDAC3 promoted myocardial growth by stimulating FGF9 and IGF2 (by repressing miR322 or miR503), which linked the signaling pathway to congenital heart disease (Jang et al, 2022, *Circ Res*, 131(2): 151-164). Here the authors described that endo HDAC3 is required for cardiomyocyte proliferation and myocardial trabeculation, which was mediated by the TGF3b signal pathway. However, the manuscript lacked the key *in vivo* data to support that conclusion. It's still unclear how endo HDAC3 regulates cardiac trabeculation. Overall, the role of HDAC3 in CHD is known, and the authors have already reported that HDAC3 mediates heart development via repression of miRNA to modulate FGF9 and IGF2. In this study, they show that HDAC3 mediates endocardium development via repressing another miRNA to modulate TGFb3. Hence, the findings reported in this manuscript, although interesting, is incremental and the *in vivo* mechanism isn't yet nailed down.

Below are my comments:

1. In mice there were no noticeable phenotypes in the various endothelial cell-derived tissues (Tie2Cre-driven HDAC3 deletion), including blood vessels (Figure 1, Supplemental Figure 2, 3). However, HDAC3 deletion in cells showed a reduction of the proliferation of immortalized neonatal MCEC. There is a disconnect between *in vitro* cell-based and *in vivo* mouse studies. Therefore, immortalized MCEC in this case isn't a good cell model to predict *in vivo* function of endothelial HDAC3.

2. The *in vivo* data (Fig. 3d) showed that there was about 20% decrease of TGFb3 mRNA level in the E11.5 heart of endothelial-specific HDAC3 deletion, suggesting that cardiac endothelial cells are not the main source of TGFb3. There was no *in vivo* data of TGFb3 protein level, which is important for understanding the impact of HDAC3 deletion on this pathway. RNA-seq data from MCEC showed a reduction of multiple growth factors after HDAC3 deletion. However, most genes with reduced expression in this RNA-seq study didn't change their protein levels, and only TGFb3 protein was reduced. This showed a disconnect between mRNA changes and protein changes. Before concluding any *in vivo* regulation by HDAC3, it's important that the authors provide *in vivo* protein data, especially for TGFb3 that had only 20% decrease in mRNA after HDAC3 deletion.

3. The manuscript claimed that endothelially-secreted TGF β 3 is a key factor that promotes the proliferation of cardiomyocytes and trabeculation, which wasn't convincingly supported by the data, given that there was only 20% decrease of TGF β 3 mRNA in the heart of the mutant mice. Could TGF β 3 secreted by other cell types, including fibroblasts, immune cells or even cardiomyocytes, compensate for the loss of 20% TGF β 3 from endothelial cells lacking HDAC3? Or was the production/secretion of TGF β 3 by other cell types compromised in the mutant heart lacking endothelial HDAC3? Also, it isn't clear whether other pathways, either cell-autonomously or non-cell-autonomously controlled by HDAC3, are required for CM proliferation and trabeculation.

4. Additional ways to determine the relevance of endothelial TGF β 3 include the following experiments:

--will endo- specific TGF β 3 deletion in mice to provide definite evidence of the in vivo function of TGF β 3 in myocardial trabeculation.

-- An alternative way to support TGF β 3 function in vivo is to perform rescue experiments, using recombinant TGF β 3 protein to treat mice of endo HDAC3 deletion, or using ex vivo heart culture treated with TGF β 3 to determine the rescue of cardiac defects caused by endo HDAC3 deletion.

5. The data showed that HDAC3 knockout in vivo and MCECs up-regulated mir-129-5p, and knockdown of mir-129-5p restored TGF β 3 expression in HDAC3 KO MCECs, suggesting that HDAC3 induces TGF β 3 expression by repressing mir-129-5p. To further support this mechanism, an in vivo study of blocking mir-129-5p by antisense oligonucleotide to recover TGF β 3 expression and improve the phenotype is very helpful.

6. The manuscript also claimed that endothelial HDAC3 was required for ECM gene expression. However, most ECM proteins are produced by cardiomyocytes during embryogenesis. it's unclear in this manuscript how ECM expression regulated by endothelial HDAC3 contributes to cardiomyocyte proliferation and the mechanism of trabeculation. One example is endothelial neuregulin 1 (Nrg1), besides promoting cell proliferation, is essential for regulating cardiomyocyte delamination to initiate ventricular trabeculation. Does endothelia HDAC3 regulate Nrg1 expression (protein level) in vivo?

7. The title does not reflect the content of the manuscript (endo HDAC3 is required for cardiac trabeculation?). It is not clear what "double brake axis" is about in the title. study.

8. In Fig. 1a, the representative staining showed endo HDAC3 deletion obviously inhibited compaction thickness, but the statistics showed there is no significant change, which is confusing.

Reviewer #3 (Remarks to the Author):

The authors of this study demonstrated that genetic knockout (KO) of histone deacetylase 3 (Hdac3) in the endocardium of mice resulted in early embryo lethality and ventricular hypotrabeculation. Through single-cell RNA sequencing, they identified a significant downregulation of extracellular matrix (ECM) components in Hdac3 KO endocardial cells. The secretome derived from cultured Hdac3 KO mouse cardiac endothelial cells was found to lack TGFB3 and exhibited a significantly reduced capacity to stimulate cultured cardiomyocyte proliferation. Notably, this impaired function was remarkably rescued by the supplementation of TGFB3. Mechanistically, the researchers discovered that HDAC3 induces Tgf β 3 expression by repressing microRNA (miR)-129-5p.

Main comments: This manuscript describes the role of an epigenetic regulator, Hdac3, in the regulation of cardiomyocyte proliferation during trabeculation using a conditional inactivation model in the endothelium (Tie2-Cre). The authors have previously described the requirement of Hdac3 in other cardiac tissues.

Results. The methodology used and the quality of the data is very good in general, although some of the panels in Figure 1B should be improved: The BrdU and pH3 stainings are very difficult to see. The authors should show high power staining of both compact and trabecular layers, and quantify cardiomyocyte proliferation in both areas separately, to determine if there are differences. At this stages, proliferation occurs mostly in compact myocardium, so one would expect to see differences also in this tissue. Indeed, Figure 1A shows a thinner compact myocardium accompanied by a less complex trabecular network, suggesting that a thinner (presumably less proliferative) compact myocardium is the cause of the thinner trabeculae. This would be also the case for the Hdac3-Nfatc2Cre model, that shows also a thinner compact myocardium.

The scRNA-seq data are very nice and informative about a depletion of ECM gene expression and together with the secretome analysis point to a deficit of proliferative signals that affect the cardiomyocytes. The authors identify Tgfb3 as the protein specifically depleted upon Hdac3 abrogation in vitro, and test the proliferative effect of this growth factor in vitro. The authors suggest that cardiac endothelial HDAC3 supports CM proliferation by inducing Tgf β 3 expression. Based on their previous results with HDAC3-epicardium specific depletion, the authors explore if miRs are affected in this endothelial HDAC3 depletion model, and identify miR-129-5p as specifically upregulated in both Hdac3KO CMs and in Hdac3KO hearts . The connection between miR-129-5p knockdown and the restoration of Tgfb3 expression in Hdac3KO CMs closes the mechanistic link between Hdac3- miR-129-5p-Tgfb3.

Discussion. It is well established that during ventricular wall development the endocardium provides patterning and proliferative signals to the underlying myocardium and is crucial for ECM synthesis. The authors have previously reported the relation between epicardial HDAC3 and miRs regulation during compaction, and this report documents that this regulatory relation in the endothelium impacts on trabeculation. There is a big emphasis on the role of Tgfb signaling in promoting CM proliferation during

trabeculation (literally: “specifically, TGFb signaling plays an instrumental role in trabeculation 50-53”) and that “knockout of either TGFb2 or TGFb3 results in hypoplastic ventricular myocardium¹⁵”. The cited references do not support a specific role for these cytokines in trabeculation but rather an involvement in valvulogenesis, although their downstream effectors may impact on chamber development. Indeed, there are reports indicative of a role for Tgfb signaling in the inhibition of CM proliferation (Kodo et al. 2016, Nat Cell Biol 18, 1031). It appears that the effect of Tgfb signaling may be context -dependent or some Tgfb be redundant; without doubting the specific effect that Tgfb3 has in promoting CM proliferation in vitro, this should not be overstated.

Response to reviewer comments

Reviewer #1

In this study, the authors investigated the role of HDAC3 in the developing endocardium during early heart development. As normal development progresses, inner cardiac endothelial cells take part into trabeculation. Failure of proper ventricular trabeculation is often associated with congenital heart disease. HDAC3 in the developing epicardium plays a critical role in supporting compact myocardial wall expansion, and it is unclear whether or how HDAC3 in the developing endocardium impacts myocardial trabeculation.

To study the role of HDAC3 in the cardiac endothelium during early heart development, Hdac3 was specifically deleted in the endothelial cells using Tie2-Cre. Hdac3 KO in the developing endothelium resulted in early embryonic lethality starting at E12.5. Histological analysis in embryos at E11.5 showed myocardial defects consisting of Hdac3^{tko} trabeculae appeared to be significantly sparser and shorter as compared to the thick and long webbing like trabecular network in littermate control hearts.

RE: Thank you for your nice summary.

General comments

The article is well written, and data are well presented and explained. The phenotype of knocking out Hdac3 is also clear. In fact, the authors demonstrate that Hdac3 KO results in hypotrabeulation, and culturing E13.5 CMs with media from a Hdac3 KO mouse cardiac endothelial cells (MCECS) line was shown to block CM proliferation. Also, culturing E13.5 CMs with media supplemented with recombinant TGFB3 rescued the proliferation capacity of CMs.

In my opinion, what is probably missing, is the confirmation of the involvement of HDAC3 downstream regulators (miR-129-5p and TGFB3) in producing the Hdac3 KO phenotype. Considering that TGFB3 KO is lethal perinatally, the phenotypic effect of overexpressing miR-129-5p in endothelial cells should produce the same effect. Also, KO of TGFB3 in endothelial cells should give the same hypotrabeulation as Hdac3 KO in the developing heart. I also would expect that overexpression of miR-129-5p to result in hypotrabeulation and possibly lethality as both Hdac3 and TgfB3 KO are lethal. Also, culturing E13.5 CMs with media supplemented with media from MCECs in which miR-129-5p has been silenced should rescue the phenotype of CM proliferation.

Re: We appreciate the reviewer's suggestions and comments. Tgfβ3 flox/+ mice are available at the Jax lab by cryo recovery (<https://www.jax.org/strain/028359>). It may take up to 6 months to receive the mice (based on our own cryo recovery experience with Jax in the past), and then it will take at least another 8 months to do all the crosses and phenotypic analyses, etc. This time-consuming in vivo work on Tgfβ3 conditional KO will be interesting, but is out of scope of our current manuscript, which is focused on understanding the function of endocardial Hdac3 during heart development. Similarly, it would be very nice to see whether overexpression of miR-129-5p in the developing endocardium recapitulates our endocardial Hdac3 KO cardiac phenotypes in vivo by generating transgenic miR-129-5p mouse lines. Again, this will conceivably take a long time to generate lines and characterize the potential phenotypes. We appreciate these great suggestions and will pursue them in the future. Injection of miR-129-5p mimics into dams may not work since we will not be able to achieve specific overexpression in the endocardium (miR mimics probably will be taken up by any cells in the embryos). Since we know that the in vivo hypotrabeulation phenotype is due to decreased cardiomyocyte proliferation, as an alternative we can address their signaling relationship as we describe next.

We generated *Tgfb3* KO and miR-129-5p overexpressing endothelial cells and harvested their supernatants, which were then used to independently treat cultured E13.5 cardiomyocytes to evaluate their effects on cardiomyocyte proliferation. Indeed, cardiomyocyte proliferation was significantly decreased in both conditions (see first figure below, a composite of Supplemental Figure 11B, page 59, and Supplemental Figure 13B, page 62). In addition, to address the phenotypic effects of endothelial *Tgfb3* KO and miR-129-5p overexpression on myocardial growth, we used an ex vivo organ culture system: we isolated wildtype E11.5 hearts and cultured them with either *Tgfb3* KO MCEC or miR-129-5p-treated MCEC supernatants, then quantified trabeculation area and compact wall thickness. This strategy is powerful and specific (suggested by Reviewer 2 as well). Our data demonstrated that either *Tgfb3* KO MCEC or miR-129-5p-treated MCEC supernatants significantly reduced their potential to induce myocardial growth as compared to control MCEC supernatant treatment (second Figure below, which is Supplemental Figure 13, page 62). Note: application of all these MCEC supernatants have global effects on the whole heart growth (not just for trabecular myocardium) since they were applied to the culture medium. This is different than the in vivo environment where the endocardial secretome predominantly impacts neighboring trabecular myocardium through paracrine signaling.

Also, we performed cardiomyocyte proliferation rescue experiments, as suggested. Supernatants from miR-129-5p knockdown *Hdac3* KO MCECs significantly rescued the cardiomyocyte proliferation defects triggered by *Hdac3* KO MCEC supernatant treatment (see results below, Figure 4H, page 42). These results provide further critical evidence to support that endothelial HDAC3 regulates myocyte proliferation through repressing miR-129-5p.

Finally, the mouse cardiac endothelial cell (MCECs) helped the authors identifying the mechanism downstream Hdac3 KO. Even though, data are significantly comparable with E12.5 primary cells or hearts, the results and statistical significance observed in control and Hdac3 KO are more striking in MCECs than in primary cells or tissues. This is shown in Figure 3D and figure 4E and G.

Re: We thank the reviewer for highlighting this important point. We agree with reviewer's comments on the discrepancy on the degree of molecular changes between *in vitro* and *in vivo* data. Both Tgfβ3 and miR-129-5p are expressed in other cardiac cells including epicardial cells. Thus, their expression from these non-endothelial cells can mask the degree of differences in cardiac endothelial cells when we used the whole heart as the sample source as shown in our original manuscript. Now, we have collected E11.5 cardiac endocardial cell population by fluorescence-activated cell sorting (FACS) and performed Tgfβ3 expression analyses by qPCR. The new updated results are shown in the revised Supplemental Figure 10 (page 58) and Figure 3D (page 40) and below. These data show more consistent results between *Hdac3* deficient cardiac endothelial cells (*in vivo*) and MCECs (*in vitro*).

Major comments:

1. Tgfβ3 expression reduction in Hdac3 KO MCECs or via HDAC3 inhibitor and in E11.5 is significant. Quite different is the percentage of downregulation in Hdac3 KO and Hdac3 inhibitor between MCECs and in E11.5 hearts (Figure 3 C and D. In Figure 3 D, the KO of Tgfβ3 is 10% in E11.5 hearts in comparison to MCECs (80-90% KO). Is 10% enough to cause the phenotype of hypotrabeulation and perinatal lethality? In favour of the authors, supplementation of recombinant TGFB3 in E13.5 CMs rescues proliferation defects due to Hdac3 KO.

RE: Please see the previous response above.

2. As a point for positive control, could the authors show the expression levels of HDAC3 in the scRNA-seq data, showing the loss of HDAC3 specifically in the endocardial cell population on the same UMAP for Figure 2E – Especially those marked by Tie+ expression?

RE: Yes. Now we provide these data in Supplemental Figure 8B on page 55 (repeated in the figure below).

3. For Figure 1B, could the authors add a heatmap/clustering profile in supplementary to show specific markers used to separate out the clusters?

RE: Thank you for your comments. Supplemental Figure 6, page 52 (repeated in the figure below) shows the clustering profile of specific markers to separate clusters.

4. While the effect on TGFB3 is clear and convincing, is there a secondary independent sgRNA/siRNA strategy used to validate the phenotype of HDAC3 KO in MCECs – To account for possible off targets?

RE: This is the same set of sgRNAs that we used in a different cell type (mouse epicardial cells) in another published study in “Circulation Research” in 2022. We did not find any off-target effects at that time. Plus, Hdac3 inhibition by Hdac3 specific inhibitor RGFP966 produced the same effects: regulation of *Tgfβ3* and miR-129-5p expression. To further address this concern, we knocked down *Hdac3* by siRNA (Dharmacon, Cat # J0043553-08-0002, 5 nM for the final concentration) and obtained a very similar result: *Hdac3*

knockdown resulted in significant *Tgfb3* downregulation (Please see figure below). Thus, the effects of *Tgfb3* downregulation were not coming from off target effects.

5. It is interesting that TGFB3 can enhance proliferation in E13.5 CMs, it might be work looking at TGFB3 supplementation in adult CMs, and see if there is any effect on proliferation at the later stages.

RE: Thank you for your interesting suggestion and we agree on this possibility. However, adult CMs are notoriously known to be very hard to culture in dish. The in vitro experiments may be technically very challenging. Of note, adult hearts have very different physiological characteristics than embryonic hearts. The findings from adult hearts may not align well with this developmental study presented here. Thus, we will stay focused on this work and will test the effects of TGFβ3 on adult CM proliferation in vivo in future studies.

6. In Figure 3G, the authors suggests that HDAC3 induces TGFB3 expression through repressing miR-129-5p, using miRZip lentivirus. The modest rescue in Figure 3G may suggest that TGFB3 expression may not be entirely regulated by miR-129-5p. Might there be other miRNAs or regulators be synergistically regulating its expression? Or other possible explanations? The authors could discuss this further.

RE: We agree with your comments. We do not claim that *Tgfb3* is entirely regulated by the HDAC3-miR-129-5p axis since we have not exhausted other potential pathways. However, the modest rescue shown in the original manuscript was most likely due to the inadequate miR-129-5p knockdown achieved at that time: the expression of miR-129-5p in *Hdac3* KO MCECs after miR-129-5p knockdown was still about 3.5 folds higher than its expression in normal control MCECs. In the revised manuscript, we redid the experiments by under puromycin selection after lentiviral transfection to remove nontransfected cells (the viral construct has a puromycin resistant gene with it, so nontransfected MCECs die off over time). Now, we achieved much better rescue effects (shown below by qPCR and western blot, and in Figure 4F and 4G, page 42, instead of only qPCR in the original submission).

7. Figure 4G. Authors show a modest rescue of miR-129-5p and TGFB3 after miRZIP129 in Hdac3 KO. One possible experiment could be silencing miR-129-5p in MCECs and supplementing E13.5 CMs with supernatant to rescue CM proliferation and strengthen the downstream role of miR-129-5p.

RE: Thanks for the suggestion. We did it as suggested. Please see the response to your first point (above).

8. TGFB3 supplementation rescue CM proliferation defects due to Hdac3 KO supernatant treatment in E13.5 CMs. Would be good to measure miR-129-5p presence in supernatant. miR-129-5p should be low in presence of TGFB3 supplementation and up in presence of Hdac3 KO.

RE: Our data suggest that miR-129-5p in cardiac endothelial cells (not secreted) works downstream of HDAC3 to regulate the expression of TGFβ3 in cardiac endothelial cells, and then TGFβ3 is secreted out of endothelial cells and provides growth signals to trabecular CMs (Figure 3). miR-129-5p works upstream of TGFβ3 and represses its expression (Figure 4). We are not sure whether miR-129-5p is secreted into supernatants. Even if miR-129-5p is secreted into supernatants and up taken by CMs, its biology in CMs appeared to be separated from our main point in this manuscript: we identified the signaling axis of Hdac3-miR-129-5p-TGFβ3 in endothelial cells, but not in cardiomyocytes. We will study the function of miR-129-5p in CMs in future studies (but not here to avoid potential confusion on the current results).

9. Would be interesting to see a possible effect of miR-129-5p silencing in primary cells in vitro or in vivo. Silencing of miR-129-5p might induce proliferation in primary CMs.

RE: Thank you for your very interesting hypothesis, as mentioned above, we will test it in our future studies (it is definitely worth pursuing further). In this manuscript, we are primarily focusing on testing whether miR-129-5p works downstream of HDAC3 in endothelial cells.

10. Mechanistically, the mechanism downstream HDAC3 is explained. Inhibition or KO of HDAC3 upregulates miR-129-5p which in turn downregulates TgfB3. What's probably missing is the phenotypic effect of the downstream regulator miR-129-5p. Upregulation of miR-129-5p or KO of TGFB3 in endothelial cells should give the same hypotrabeulation as Hdac3 KO in the developing heart. Interestingly, KO of TgfB3 in embryo is lethal as well as Hdac3 KO. It would be interesting to see whether miR-129-5p upregulation is also lethal, considering its function in between Hdac3 and TgfB3.

RE: Please see our response to your general comments. Again, we would like to perform these interesting studies in the future from different perspectives. In the current report, we are focused on the signaling axis in term of gene regulation. We believe both miR-129-5p and TGFβ3 may play important and exclusive roles on the context how endocardial HDAC3 regulates myocardial trabeculation. HDAC3 also regulates ECMs as shown in this report, which very likely contribute to this process as well.

11. The authors should also define what they mean by “double brake axis” in their title, as there were no reference or explanation to what it means in the main text.

RE: Thanks for picking up the misleading title. Now we change it to “endocardial HDAC3 is required for myocardial trabeculation”.

Minor comments

1. Figure 4. Typo at lines 744 745 746. It is the quantification of miR-129-5p and not TgfB3.

RE: Thanks. We corrected it.

Reviewer #2

The manuscript by Jang et al reports a new signaling pathway mediated by HDAC3 that is highly relevant to congenital heart disease (CHD). Epigenetic regulation of mammalian development is well studied. HDAC3 has been linked to embryogenesis and multiple organ development by several studies (Reviewed in detail by Dsilva et al in Mol Reprod Dev, 2022, 90(1): 14-26). For example, HDAC3 was shown to physically interact with Tbx5 to regulate Tbx5 acetylation, which represses Tbx5-dependent expression of cardiac lineage-specific genes, supporting a direct role of HDAC3 in early cardiogenesis (Lewandowski et al, 2014, Human Mol Genet, 23(14):3801-3809). The authors previously demonstrated a signaling pathway through which HDAC3 promoted myocardial growth by stimulating FGF9 and IGF2 (by repressing miR322 or miR503), which linked the signaling pathway to congenital heart disease (Jang et al, 2022, Circ Res, 131(2): 151-164). Here the authors described that endo HDAC3 is required for cardiomyocyte proliferation and myocardial trabeculation, which was mediated by the TGF3b signal pathway. However, the manuscript lacked the key in vivo data to support that conclusion. It's still unclear how endo HDAC3 regulates cardiac trabeculation. Overall, the role of HDAC3 in CHD is known, and the authors have already reported that HDAC3 mediates heart development via repression of miRNA to modulate FGF9 and IGF2. In this study, they show that HDAC3 mediates endocardium development via repressing another miRNA to modulate TGFb3. Hence, the findings reported in this manuscript, although interesting, is incremental and the in vivo mechanism isn't yet nailed down.

RE: We totally agree with and appreciate your overall comments. As we stated in the "Introduction", HDAC3 plays critical roles during heart development in general, but its biological functions in different types of cardiac cells are quite divergent, in a more cell-context dependent manner. We have added a citation to the recent HDAC review paper (Dsilva P et al, Mol Reprod Dev. 2023, 90:14-26) in our revised manuscript (reference 20, page 4) to acknowledge the current existing findings/understandings. Whether and how endocardial HDAC3 regulates ventricular trabeculation (which is very different than compaction in many ways) is unknown until we reported our findings here. We acknowledge that the general working mechanisms are similar to what we discovered in our previous epicardium-myocardial compaction story: HDAC3-microRNA-growth factor, although HDAC3 works with different microRNA and signaling pathway in this unique novel developmental module as to how ventricular trabeculation is formed. This would be an important new knowledge to the trabeculation developmental biology. on the other hand, the overall similar mechanistic findings from two completely different myocardial morphogenic processes will reinforce that HDAC3 in the nonmyocardial compartments has profound impact on myocardial development.

Below are my comments:

1. In mice there were no noticeable phenotypes in the various endothelial cell-derived tissues (Tie2Cre-driven HDAC3 deletion), including blood vessels (Figure 1, Supplemental Figure 2, 3). However, HDAC3 deletion in cells showed a reduction of the proliferation of immortalized neonatal MCEC. There is a disconnect between in vitro cell-based and in vivo mouse studies. Therefore, immortalized MCEC in this case isn't a good cell model to predict in vivo function of endothelial HDAC3.

RE: We acknowledge fundamental differences between MCECs and in vivo embryonic cardiac endothelial cells. For instance, MCECs are engineered and immortalized, so they grow quickly under culture. Thus, the proliferation difference can be relatively easier to identify in MCECs. In vivo endothelial cells won't grow that quickly and there is not enough time to see the proliferation difference given *Hdac3* KO embryos die sooner (most die at

E12.5). We use MCECs mainly to study their paracrine signaling mechanisms, and then we validate the findings in vivo. Thus, we think these MCECs are suitable for this purpose.

2. The in vivo data (Fig. 3d) showed that there was about 20% decrease of TGFb3 mRNA level in the E11.5 heart of endothelial-specific HDAC3 deletion, suggesting that cardiac endothelial cells are not the main source of TGFb3. There was no in vivo data of TGFb3 protein level, which is important for understanding the impact of HDAC3 deletion on this pathway. RNA-seq data from MCEC showed a reduction of multiple growth factors after HDAC3 deletion. However, most genes with reduced expression in this RNA-seq study didn't change their protein levels, and only TGF3b protein was reduced. This showed a disconnect between mRNA changes and protein changes. Before concluding any in vivo regulation by HDAC3, it's important that the authors provide in vivo protein data, especially for TGF3b that had only 20% decrease in mRNA after HDAC3 deletion.

RE: We thank the reviewer for the constructive comments and suggestions. Now we performed TGFβ3 western blot on E11.5 *Hdac3^{tko}* hearts. We found that TGFβ3 is significantly decreased in *Hdac3^{tko}* hearts (please see figure below and Figure 3E on page 40). We also tried to perform Immunofluorescence staining on E11.5 hearts with several TGFβ3 antibodies. Unfortunately, none of them worked. Yes, TGFβ3 is also expressed in other cardiac cells such as epicardial cells and cushion mesenchymal cells [PMID #: 10191064, 12815630]. However, TGFβ3 from these cells will stay locally but won't traffic far to interact with trabecular myocardium (this is true for many other growth factors such as Nrg1, FGFs, IGFs which effect mostly by paracrine fashion at the anatomical proximity during heart development). To clearly demonstrate the downregulation of Tgfβ3 mRNA levels in *Hdac3* KO cardiac endothelial cells, we sorted Tie2-derived cells out first and then performed qPCR (please see the first response to reviewer #1 and the result is also shown on Figure 3E page 40). Within Tie2-derived cell population, Tgfβ3 was significantly downregulated in *Hdac3^{tko}* hearts.

3. The manuscript claimed that endothelially-secreted TGFb3 is a key factor that promotes the proliferation of cardiomyocytes and trabeculation, which wasn't convincingly supported by the data, given that there was only 20% decrease of TGF3b mRNA in the heart of the mutant mice. Could TGF3b secreted by other cell types, including fibroblasts, immune cells or even cardiomyocytes, compensate for the loss of 20% TGF3b from endothelial cells lacking HDAC3? Or was the production/secretion of TGF3b by other cell types compromised in the mutant heart lacking endothelial HDAC3? Also, it isn't clear whether other pathways, either cell-autonomously or non-cell-autonomously controlled by HDAC3, are required for CM proliferation and trabeculation.

RE: The trabeculation phenotypes in cardiomyocytes are non-cell-autonomous, since HDAC3 expression in cardiomyocytes are still intact. Yes, TGFβ3 is likely expressed in

many cardiac cells including epicardial cells and cushion mesenchymal cells, that also likely explained why we did not see a greater reduction of *Tgfβ3* mRNA levels from the whole heart lysates (although that data is the overall mRNA levels, we did see a larger decrease of TGFβ3 protein levels, as shown above). Trabeculation promoting factors from nonmyocytes are mainly contributed by endocardial cells through paracrine signaling effects due to their anatomical proximity. TGFβ3 from cushion mesenchymal cells or epicardial cells will not be able to traffic that far to support trabecular myocardial growth. At E11.5 or early, cardiac fibroblasts have not been generated since endoMT has not occurred yet.

4. Additional ways to determine the relevance of endothelial TGFβ3 include the following experiments:

--will endo- specific TGFβ3 deletion in mice to provide definite evidence of the in vivo function of TGFβ3 in myocardial trabeculation.

RE: Thank you for your suggestion, please see the first response to reviewer #1.

-- An alternative way to support TGFβ3 function in vivo is to perform rescue experiments, using recombinant TGFβ3 protein to treat mice of endo HDAC3 deletion, or using ex vivo heart culture treated with TGFβ3 to determine the rescue of cardiac defects caused by endo HDAC3 deletion.

RE: injection of recombinant TGFβ3 proteins to dams are very likely unable to generate specific effects on trabecular myocardium since the whole body of embryos can absorb it, so it will be very difficult to interpret the results. Meanwhile, we took your great suggestion of ex vivo organ culture experiments. We found that treatment of TGFβ3 recombinant protein significantly rescued the hypotrabeculation phenotypes in *Hdac3^{tko}* hearts (please see figure below and Supplemental Figure 12 on page 61).

5. The data showed that HDAC3 knockout in vivo and MCECs up-regulated mir-129-5p, and knockdown of mir-129-5p restored TGFb3 expression in HDAC3 KO MCECs, suggesting that HDAC3 induces TGFb3 expression by repressing mir-129-5p. To further support this mechanism, an in vivo study of blocking mir-129-5p by antisense oligonucleotide to recover TGFb3 expression and improve the phenotype is very helpful.

RE: Thank you for the suggestion. We think anti-miR-129-5p treatment cannot achieve endothelial specificity since miR-129-5p is expressed in many cell types including cardiomyocytes (PMID #: 31815867; 26946427).

6. The manuscript also claimed that endothelial HDAC3 was required for ECM gene expression. However, most ECM proteins are produced by cardiomyocytes during embryogenesis. it's unclear in this manuscript how ECM expression regulated by endothelial HDAC3 contributes to cardiomyocyte proliferation and the mechanism of trabeculation. One example is endothelial neuregulin 1 (Nrg1), besides promoting cell proliferation, is essential for regulating cardiomyocyte delamination to initiate ventricular trabeculation. Does endothelia HDAC3 regulate Nrg1 expression (protein level) in vivo?

RE: Mesenchymal cells in the developing heart are the main source of secreting ECM, endothelial cells are also a major ECM source before some of them undergo endoMT to turn into mesenchymal cells (PMID #: 21618406). The ECM between endocardium and trabeculation (also called cardiac jelly) is known to be important for providing a media for endocardial growth signals to traffic to the growing trabecular myocardium (PMID #: 23843320; 29536133). The compromise of these ECM will affect trabecular formation. We did check the expression of NRG1 in vivo, but we did not find significant difference between *Hdac3^{tko}* hearts and control hearts (Please see figure below).

7. The title does not reflect the content of the manuscript (endo HDAC3 is required for cardiac trabeculation?). It is not clear what “double brake axis” is about in the title. study.

RE: We thank you for your suggestion and have changed it to: Endocardial Hdac3 is required for myocardial trabeculation.

8. In Fig. 1a, the representative staining showed endo HDAC3 deletion obviously inhibited compaction thickness, but the statistics showed there is no significant change, which is confusing.

RE: Yes, overall, the compact wall thickness is thinner in Hdac3 KO hearts, but did not reach statistical significance in our original results (p=0.175), likely due to low sample number. Now we have obtained a few more samples (n=7 for E12.5 *Hdac3^{tko}* hearts and n=8 for E12.5 CTL hearts) and redid the statistical analysis. With the additional samples analyzed, both trabeculae area and compact wall thickness were statistically significantly

reduced for *Hdac3^{tko}* hearts (the new quantification graphs shown below and in Figure 1A on page 36).

Reviewer #3

Main comments: This manuscript describes the role of an epigenetic regulator, Hdac3, in the regulation of cardiomyocyte proliferation during trabeculation using a conditional inactivation model in the endothelium (Tie2-Cre). The authors have previously described the requirement of Hdac3 in other cardiac tissues.

Results. The methodology used and the quality of the data is very good in general, although some of the panels in Figure 1B should be improved: The BrdU and pH3 stainings are very difficult to see. The authors should show high power staining of both compact and trabecular layers, and quantify cardiomyocyte proliferation in both areas separately, to determine if there are differences. At this stages, proliferation occurs mostly in compact myocardium, so one would expect to see differences also in this tissue. Indeed, Figure 1A shows a thinner compact myocardium accompanied by a less complex trabecular network, suggesting that a thinner (presumably less proliferative) compact myocardium is the cause of the thinner trabeculae. This would be also the case for the Hdac3-Nfatc2Cre model, that shows also a thinner compact myocardium.

RE: Thank you for the insightful comments and suggestions. In the original manuscript, data for % p-H3+ CMs and % BrdU+ CMs were presented with respect to quantification of the trabecular zone; we now more fully describe those results. As suggested, we have now also quantified the CM proliferation index in the compact zone (these new results are shown below and on Figure 1B, page 36). Indeed, the proliferation index is significantly lower in *Hdac3^{tko}* hearts as compared to control hearts. Now we have also magnified a small region of compact layer to show high power images for better visibility (we apologize for being unable to magnify the whole picture due to figure size limits).

The scRNA-seq data are very nice and informative about a depletion of ECM gene expression and together with the secretome analysis point to a deficit of proliferative signals that affect the cardiomyocytes. The authors identify Tgfb3 as the protein specifically depleted upon Hdac3 abrogation in vitro, and test the proliferative effect of this growth factor in vitro. The authors suggest that cardiac endothelial HDAC3 supports CM proliferation by inducing Tgfβ3 expression. Based on their previous results with HDAC3-epicardium specific depletion, the authors explore if miRs are affected in this endothelial HDAC3 depletion model, and identify miR-129-5p as specifically upregulated in both Hdac3KO CMs and in Hdac3KO hearts. The connection between miR-129-5p knockdown and the restoration of Tgfb3 expression in Hdac3KO CMs closes the mechanistic link between Hdac3- miR-129-5p-Tgfb3.

RE: Thank you for the nice summary.

Discussion. It is well established that during ventricular wall development the endocardium provides patterning and proliferative signals to the underlying myocardium and is crucial for ECM synthesis. The authors have previously reported the relation between epicardial HDAC3 and miRs regulation during compaction, and this report documents that this regulatory relation in the endothelium impacts on trabeculation. There is a big emphasis on the role of Tgfb signaling in promoting CM proliferation during trabeculation (literally: “specifically, TGFb signaling plays an instrumental role in trabeculation”) and that “knockout of either TGFb2 or TGFb3 results in hypoplastic ventricular myocardium”. The cited references do not support a specific role for these cytokines in trabeculation but rather an involvement in valvulogenesis, although their downstream effectors may impact on chamber development. Indeed, there are reports indicative of a role for Tgfb signaling in the inhibition of CM proliferation (Kodo et al. 2016, Nat Cell Biol 18, 1031). It appears that the effect of Tgfb signaling may be context -dependent or some Tgfb be redundant; without doubting the specific effect that Tgfb3 has in promoting CM proliferation in vitro, this should not be overstated.

RE: We completely agree with your comments. We agree that the impact of TGFβ signaling on cardiomyocyte proliferation is highly likely context dependent. We have added some additional discussion and edited some wording to avoid any potential inaccurate/over statement. In our review of the cited references, their histology data did show clear hypoplastic myocardia in the KO hearts, which suggests that TGFβ signaling promotes myocardial growth.

REVIEWER COMMENTS

Reviewer #1 (Remarks to the Author):

The authors have substantially revised the manuscript with additional experiments using TGF β 3 KO and miR-129-5p overexpressing endothelial cells, to support the role of HDAC3 in regulating myocyte proliferation via miR-129-5p. Additionally, using E11.5 Cardiac explants and FACS-sorting of endocardial cells, the authors have clarified the previous difference in effect sizes, and majority of my comments.

While the authors have shown evidence regulated by the HDAC3-miR-129-5p axis, at least other potential signaling pathways, its linkages and future research gap should be discussed in the discussion section.

Below highlighted are some revisions that I think are reasonable and within an appropriate scope of this manuscript – taking into consideration comments from Reviewer #2 :

Reviewer #2 outlines the concern in Figure 2E relating to the Tie2-Cre activity in both endothelial cells and fibroblasts, which makes assessable DEG difficult. Nonetheless, despite the Tie2-specific deletion of Hdac3 in some fibroblast cells, there is a clearer separation of cluster between control vs. Hdac3 ko in the CEC population relative to the FC1/FC2 population.

The percentage of cells expressing Cre in sub-populations might be clearer to address this question from Reviewer #2. With scRNA-seq, the DEGs can be teased out separately amongst the CEC clusters, which seem to have a stronger effect, as expected, on the endothelial population.

The technical concerns with regard to the source of endothelial cells used (from Reviewer #2)- in this case cultured immortalized mouse cardiac endothelial cells (MCEC) to primary cardiac endothelial cells (pEC). Primary cardiac endothelial cells may be preferred but will be technically challenging for genetic manipulation – as a result, it is not so feasible. Instead, it is acceptable to use MCEC as long as there are proper controls.

While it is ideal to have a mouse model of endothelial-specific TGF β 3 deletion (also in my previous comment), it is certainly true that the purchase and breeding of TGF β 3 flox/+ mice will be time-consuming, beyond the scope of this submission. Additionally, despite the use of MCECs, it is impressive that the removal of TGF β 3 and supplemental miR-129-5p mimics in MCECs supernatant, shown in Supplemental Figure 11 and 13, there is some reduction in trabeculation area in E11.5 Wild-type cardiac explants, and in Supplemental Figure 12 (provided in the revised manuscript) shows that TGF β 3 supplementation alone could rescue the trabeculation defect seen in endocardial Hdac3 KO within 24 hours, implicating the importance of TGF β 3 in Tie2-specific deletion mice model. **It is, therefore, crucial also to clarify if myocardial**

growth in endocardial Hdac3 deficient hearts can be rescued somewhat when using supernatant Wild-type MCECs (EV) control, but not with TGFβ3 deletion, which may have been excluded in the figures.

As per the final comment by Reviewer #2, as Hdac3 is expected to deacetylate other downstream targets, the Tie2-specific Cre-mediated deletion of endothelial Hdac3 may have other pleiotropic effects implicating other regulatory pathways involved in cardiac development apart from TGFβ3. As mentioned in my comments, **while it is beyond the scope of this present study, it should be discussed under the discussion section.**

Reviewer #2 (Remarks to the Author):

The focus of this manuscript is endothelial-HDAC3 mediated cardiomyocyte proliferation and myocardial trabeculation. In both the Tie2-Cre mediated HDAC3 KO and the control heart, there was significant number of fibroblasts that showed Cre activity (Fig 2E), suggesting a role of fibroblastic HDAC3 in cardiomyocyte morphogenesis. It's therefore not clear the relative contribution of fibroblasts vs endothelial cells to the process of HDAC3-mediated myocyte proliferation and trabeculation. It is difficult to know what level of assessable DEG makes biological significance from scRNA-Seq. Therefore, data derived from the in vitro model is very important for this study. The MCEC line (CEDARLANE Labs Cat. No. CLU510 and Ref 37) used in this study was immortalized by co-expression of hTERT and SV40-LT. Since the role of paracrine signaling is the main finding of this study, it is important to answer how the secretome of primary MCEC is different from the immortalized cell line used in this study. This will answer whether the in vitro model used in this study validates the conclusion made by the authors. Key experiments should have been repeated using primary cardiac endothelial cells. Overall, my previous concern about the disconnect between in vitro cell based and in vivo mouse studies is still not addressed in this revised submission.

In TGFb3 germline deletion mice, approximately 26% of fetuses showed myocardial defects affecting compaction and trabeculation, suggesting TGFb3 is involved in ventricular myocardium development (DOI: 10.3390/jcdd7020019). The manuscript didn't show enough evidence to support that endothelial-derived TGFb3 is required for myocardium compaction and trabeculation given that TGFb3 is a secreted factor and that there is only 50% decrease of cardiac TGFb3 in the endothelial-specific HDAC3 KO model (The rebuttal data showed 50% decrease of TGFb3 protein). So, the data from the mouse model of endothelial-specific TGFb3 deletion is needed to determine that endothelial TGFb3 is required for cardiac compaction and trabeculation.

Compared to TGFb3 germline deletion that caused newborn mice dead within 24h of birth (<https://doi.org/10.1038/ng1295-415>; <https://doi.org/10.1038/ng1295-409>), Tie2-Cre mediated endothelial HDAC3 deletion caused death of almost all fetuses at E14.5 (Supplemental data Table 1), showing more severe mortality but with only 50% decrease of TGFb3 protein. Based on the data from these two mouse models, the concern is whether other key factors or pathways involved in cardiac development are regulated by endothelial HDAC3 besides TGFb3 pathway.

Even with the newly added data in this revised manuscript, key data required to validate the role of endothelial vs other cells HDAC3 are still missing.

Reviewer #3 (Remarks to the Author):

The authors have answered my comments and suggestions in a satisfactory manner and added in vitro mechanistic insights.

Response to reviewer comments

Reviewer #1 (Remarks to the Author):

The authors have substantially revised the manuscript with additional experiments using TGFB3 KO and miR-129-5p overexpressing endothelial cells, to support the role of HDAC3 in regulating myocyte proliferation via miR-129-5p. Additionally, using E11.5 Cardiac explants and FACS-sorting of endocardial cells, the authors have clarified the previous difference in effect sizes, and majority of my comments.

While the authors have shown evidence regulated by the HDAC3-miR-129-5p axis, at least other potential signaling pathways, its linkages and future research gap should be discussed in the discussion section.

Re: Thank you very much for your continuous support on improving our manuscript. We agree that discussion of other signaling pathways and future research gap is needed. Please refer to the new discussion paragraph added on page 18 and we also pasted it below.

“On the other hand, HDAC3, as a histone modifier, may have many downstream targets that participate in other signaling pathways and exert various other biological effects. For instance, several Wnt ligands (e.g., *Wnt4*, *Wnt5a*, *Wnt11*) and Hedgehog signaling components (*Smo*, *EVC*, *Gli2*) were also significantly downregulated in *Hdac3* KO MCECs. The disruption of these signaling pathways may affect cell adhesion, migration, and ciliogenesis, thus, also contributing to the overall *Hdac3*^{tko} cardiac phenotypes. We will be actively investigating these possibilities in our future studies.”

The comments on R2's critics:

Below highlighted in red are some revisions that I think are reasonable and within an appropriate scope of this manuscript – taking into consideration comments from Reviewer #2 :

Reviewer #2 outlines the concern in Figure 2E relating to the Tie2-Cre activity in both endothelial cells and fibroblasts, which makes assessable DEG difficult. Nonetheless, despite the Tie2-specific deletion of *Hdac3* in some fibroblast cells, there is a clearer separation of cluster between control vs. *Hdac3* ko in the CEC population relative to the FC1/FC2 population. **The percentage of cells expressing Cre in sub-populations might be clearer to address this question from Reviewer #2.** With scRNA-seq, the DEGs can be teased out separately amongst the CEC clusters, which seem to have a stronger effect, as expected, on the endothelial population.

Re: Thank you for the guidance in addressing R2's concerns. As suggested, by leveraging both scRNA-Seq on YFP+ cells and Tie2 fate mapping immunostaining data, we demonstrate that Tie2-Cre is predominantly expressed in cardiac endothelial cell (CEC) population but not in mesenchymal cells (MC1) and mesenchymal cells (MC2). MC1 and MC2 are derived from early Tie2+ endothelial progenitors through endothelial-mesenchymal transition (EndoMT) but losing expression of endothelial specific genes (e.g., *Tie1*, *Cdh5*) afterwards. Note, MC1 and MC2 are physically located in the AV cushion area and these cells are predominantly engaged in future valve formation and membranous inter-chamber septum formation (PMID: 7034167; 29787124), not having known functions in myocardial trabeculation as they are physically far away from trabecular myocardium. CECs do contribute to the cardiac fibroblast population during heart development, although these fibroblasts are not even formed before E12.5 (PMID: 31570334; 26748307). We performed scRNA-Seq at E11.5. To avoid confusion, we rename “FC1” and “FC2” to “MC1” and “MC2”, respectively. Please refer to the figure below for data details. We hope this adequately addresses R2's concerns.

The technical concerns with regard to the source of endothelial cells used (from Reviewer #2)- in this case cultured immortalized mouse cardiac endothelial cells (MCEC) to primary cardiac endothelial cells (pEC). Primary cardiac endothelial cells may be preferred but will be technically challenging for genetic manipulation – as a result, it is not so feasible. Instead, it is acceptable to use MCEC as long as there are proper controls.

While it is ideal to have a mouse model of endothelial-specific TGF β 3 deletion (also in my previous comment), it is certainly true that the purchase and breeding of TGF β 3 flox/+ mice will be time-consuming, beyond the scope of this submission. Additionally, despite the use of MCECs, it is impressive that the removal of TGF β 3 and supplemental miR-129-5p mimics in MCECs supernatant, shown in Supplemental Figure 11 and 13, there is some reduction in trabeculation

area in E11.5 Wild-type cardiac explants, and in Supplemental Figure 12 (provided in the revised manuscript) shows that TGF β 3 supplementation alone could rescue the trabeculation defect seen in endocardial Hdac3 KO within 24 hours, implicating the importance of TGF β 3 in Tie2-specific deletion mice model. **It is, therefore, crucial also to clarify if myocardial growth in endocardial Hdac3 deficient hearts can be rescued somewhat when using supernatant Wild-type MCECs (EV) control, but not with TGF β 3 deletion, which may have been excluded in the figures.**

Re: Thank you for your additional suggestion on myocardial growth rescue using wildtype MCEC supernatants. We performed these experiments and have provided an additional supplemental figure (Supplemental Fig. 11 on page 60, and pasted below). Wildtype MCEC supernatants successfully rescued the hypotrabeulation phenotype seen on Hdac3^{tko} hearts.

As per the final comment by Reviewer #2, as Hdac3 is expected to deacetylate other downstream targets, the Tie2-specific Cre-mediated deletion of endothelial Hdac3 may have other pleiotropic effects implicating other regulatory pathways involved in cardiac development apart from TGF β 3. As mentioned in my comments, **while it is beyond the scope of this present study, it should be discussed under the discussion section.**

Re: We agree with this point and added a new paragraph of discussion. Please see the response to R1's 1st criticism.

Reviewer #2 (Remarks to the Author):

The focus of this manuscript is endothelial-HDAC3 mediated cardiomyocyte proliferation and myocardial trabeculation. In both the Tie2-Cre mediated HDAC3 KO and the control heart, there was significant number of fibroblasts that showed Cre activity (Fig 2E), suggesting a role of fibroblastic HDAC3 in cardiomyocyte morphogenesis. It's therefore not clear the relative contribution of fibroblasts vs endothelial cells to the process of HDAC3-mediated myocyte proliferation and trabeculation. It is difficult to know what level of assessable DEG makes biological significance from scRNA-Seq. Therefore, data derived from the in vitro model is very important for this study. The MCEC line (CEDARLANE Labs Cat. No. CLU510 and Ref 37) used in this study was immortalized by co-expression of hTERT and SV40-LT. Since the role of paracrine signaling is the main finding of this study, it is important to answer how the secretome of primary MCEC is different from the immortalized cell line used in this study. This will answer whether the in vitro model used in this study validates the conclusion made by the authors. Key experiments should have been repeated using primary cardiac endothelial cells. Overall, my previous concern about the disconnect between in vitro cell based and in vivo mouse studies is still not addressed in this revised submission.

In TGFb3 germline deletion mice, approximately 26% of fetuses showed myocardial defects affecting compaction and trabeculation, suggesting TGFb3 is involved in ventricular myocardium development (DOI: 10.3390/jcdd7020019). The manuscript didn't show enough evidence to support that endothelial-derived TGFb3 is required for myocardium compaction and trabeculation given that TGFb3 is a secreted factor and that there is only 50% decrease of cardiac TGFb3 in the endothelial-specific HDAC3 KO model (The rebuttal data showed 50% decrease of TGFb3 protein). So, the data from the mouse model of endothelial-specific TGFb3 deletion is needed to determine that endothelial TGFb3 is required for cardiac compaction and trabeculation.

Compared to TGFb3 germline deletion that caused newborn mice dead within 24h of birth (<https://doi.org/10.1038/ng1295-415>; <https://doi.org/10.1038/ng1295-409>), Tie2-Cre mediated endothelial HDAC3 deletion caused death of almost all fetuses at E14.5 (Supplemental data Table 1), showing more severe mortality but with only 50% decrease of TGFb3 protein. Based on the data from these two mouse models, the concern is whether other key factors or pathways involved in cardiac development are regulated by endothelial HDAC3 besides TGFb3 pathway.

Even with the newly added data in this revised manuscript, key data required to validate the role of endothelial vs other cells HDAC3 are still missing.

Re: Thank you for your critique.

Regarding the potential role of fibroblasts in contribution to the hypotrabeulation phenotype, Figure 2E only showed that these “fibroblasts” (technically they are not fibroblasts as clarified above. We are sorry for the confusion of our nomenclature in our early versions of the manuscript) are derived from early *Tie2+* progenitor cells through endothelial-mesenchymal transition (EndoMT), but they are not Cre positive (Please see the 2nd response to R1's comments). There are no cardiac fibroblasts from endoMT before E12.5, while the hypotrabeulation phenotype in *Hdac3^{tko}* hearts is already apparent at E11.5.

Regarding using MCEC cells, as many would agree, it is technically quite challenging to acquire a large number of primary cardiac endothelial cells from E11.5 hearts in addition to the difficulty of having a large enough number of *Hdac3^{tko}* hearts in the same batch. We initially attempted this approach, but were unable to overcome these technical hurdles, thus we chose to use MCEC cells as a substitute. We acknowledge that there is a difference

and a limitation using an *in vitro* immortalized cell line versus primary cells. However, depending on how they are used, MCECs are still an important research tool and are widely used by many other scientists. Importantly, we have validated most findings from MCECs in the *in vivo* animals throughout the whole study as shown in our manuscript.

Regarding the experiment with endothelial conditional knockout of *Tgfβ3* as suggested in the 1st round, we have responded accordingly. We will try this experiment in the future.

Regarding the phenotypic discrepancy between *Hdac3* endothelial knockouts and *Tgfβ3* global knockouts, we acknowledge that there are phenotypic differences between these two different knockout models, as we know that both proteins have broad downstream targets that can impact on many biological functions, but not necessarily redundant to each other. Thus, it is reasonable to see the phenotypic difference between them. Note: we do not claim that HDAC3 regulates trabeculation exclusively through TGFβ3, but rather demonstrate that this axis contributes to trabeculation development. We concur that there might be other non-TGFβ3 pathways that contribute to *Hdac3^{tko}* cardiac phenotype, and we added a new paragraph of discussion (page 18). R1 also had the same comment. Thanks.

Reviewer #3 (Remarks to the Author):

The authors have answered my comments and suggestions in a satisfactory manner and added *in vitro* mechanistic insights.

Re: Thank you for your support on improving our manuscript.

REVIEWERS' COMMENTS

Reviewer #1 (Remarks to the Author):

The authors have addressed concerns raised